

**Does the GPM mission improve the systematic error component in satellite**
**rainfall estimates over TRMM, an evaluation at a pan-India scale?**
Harsh Beria[1], Trushnamayee Nanda[1], Deepak Singh Bisht[1], Chandranath Chatterjee[1]
[1]Agricultural and Food Engineering Department, Indian Institute of Technology Kharagpur,
Kharagpur, India
*Correspondence to:* Harsh Beria (harsh.beria93@gmail.com)
**Abstract.** Last couple of decades have seen the outburst of a number of satellite based
precipitation products with Tropical Rainfall Measuring Mission (TRMM) as the most widely
used for hydrologic applications. Transition of TRMM into Global Precipitation Mission
(GPM) promises enhanced spatio-temporal resolution along with upgrades in sensors and
rainfall estimation techniques. Dependence of systematic error components in rainfall
estimates of Integrated Multi-satellitE Retrievals for GPM (IMERG), and their variation with
climatology and topography, was evaluated over 86 basins in India for year 2014 and
compared with the corresponding (2014) and retrospective (1998-2013) TRMM estimates.
IMERG outperformed TRMM for all rainfall intensities across a majority of Indian basins,
with significant improvement in low rainfall estimates showing smaller negative biases in 75
out of 86 basins. IMERG increased the inter-basin variability in bias for medium and high
rainfall estimates. Low rainfall estimates in TRMM showed a systematic dependence on
basin climatology, with significant overprediction in semi-arid basins which gradually
improved in the higher rainfall basins. Medium and high rainfall estimates of TRMM
exhibited a strong dependence on basin topography, with declining skill in the higher
elevation basins. Systematic dependence of error components on basin climatology and
topography was reduced in IMERG, especially in terms of topography. Rainfall-runoff
modeling using Variable Infiltration Capacity (VIC) model over a flood prone basin
(Mahanadi) revealed that improvement in rainfall estimates in IMERG didn't translate into
improvement in runoff simulations. More studies are required over basins in different hydro-
climatic zones to evaluate the hydrologic significance of IMERG.
**Keywords:** GPM, IMERG, TRMM, VIC, climatology, topography





## 1 Introduction

The developing part of the world suffers from acute data shortage, both in terms of
quality and quantity. A recent commentary from Mujumdar (2015) provided insights into the
problems faced by the Indian hydrologic community due to the lack of willingness of the
relevant governmental bodies to openly share meteorologic and hydrologic data and its meta
data to the research community. With the threats of climate changing looming large, high
quality precipitation products (in terms of accuracy, spatial and temporal resolution) are the
need of the hour. Satellite precipitation products offer a viable alternative to gauge based
rainfall estimates.
A number of satellite based precipitation estimates have cropped up in the past two
decades, the famous ones being Climate Prediction Center morphing technique (CMORPH),
Precipitation Estimation from Remotely Sensed Information Using Artificial Neural
Networks (PERSIANN), PERSIANN Climate Data Record (PERSIANN-CDR), Tropical
Rainfall Measuring Mission (TRMM), Asian Precipitation - Highly-Resolved Observational
Data Integration Towards Evaluation (APHRODITE) and National Oceanic and Atmospheric
Administration (NOAA) Climate Prediction Center (CPC). A number of studies over the past
decade have evaluated the hydrologic application of these datasets over regions with varied
topography and climatology.
Artan et al. (2007) used CPC to drive a hydrologic model over four basins with varied
hydro-climatic and physiographic conditions in Africa and South-east Asia and reported
similar rainfall-runoff performance on calibration using gauge and satellite rainfall estimates.
Collischonn et al. (2008) also reported reasonable streamflow simulations using TRMM
estimates over an Amazon River basin. Akhtar et al. (2009) used multiple artificial neural
networks (ANN) to forecast discharges at varying lead times using TRMM 3B42V6
precipitation estimates. Wu et al. (2012) used TRMM 3B42V6 estimates to develop a real-
time flood monitoring system and concluded that the probability of detection (POD)
improved with longer flood durations and larger affected areas. Kneis et al. (2014) evaluated
TRMM 3B42-V7 and its real-time counterpart TRMM 3B42-V7RT over Mahanadi River
basin in India and found the research product (3B42) to be superior to the real-time
alternative (3B42RT) in terms of both the statistical and hydrologic components. Peng et al.
(2014) found a systematic dependence of TRMM estimates on climatology in North-West
China, characterizing the wetter regions better than the drier conditions. They also reported



promising results in the streamflow simulations at ungauged basin in arid and semi-arid
regions.  Bajracharya et al. (2014) used CPC to drive a hydrologic model over Bagmati basin
in Nepal and reported that the incorporation of local rain gauge data in addition to CPC
tremendously benefited the streamflow simulations. Shah and Mishra (2015) explored the
uncertainty in the estimates of multiple satellite rainfall products over major Indian basins
and investigated the influence of bias in the satellite rainfall products on flood simulation
over Mahanadi River basin in India. Most of the studies which evaluated multiple satellite
precipitation estimates have reported TRMM to give the best estimate over the Tropical part
of the world (Gao and Liu, 2013; Prakash et al., 2016b; Zhu et al., 2016).
Tropical Rainfall Measuring Mission (TRMM) satellite was launched in late 1997 and
provides high resolution (0.25° x 0.25°) quasi-global (50° N-S) rainfall estimates (Huffman et
al., 2007).  The TRMM mission is a joint mission between the National Aeronautics and
Space Administration (NASA) and the Japan Aerospace Exploration (JAXA) Agency to
study rainfall for weather and climate research. The TRMM satellite produced 17 years of
valuable precipitation data over the Tropics. In the last decade, a number of studies have
evaluated Tropical Rainfall Measuring Mission (TRMM) Multi-Resolution Analysis (TMPA)
product over different topographies and climatologies.
Owing to the tremendous success of TMPA mission, Global Precipitation
Measurement (GPM) was launched on February 27, 2014 (Liu, 2016). The GPM sensors
carry first spaceborne dual-frequency phased array precipitation radar (DPR) operating at Ku
(13 GHz) and Ka (35 GHz) bands and a canonical-scanning multichannel (10-183 GHz)
microwave imager (GMI) (Hou et al., 2014). The improved sensitivity of Ku and Ka bands
allow for improved detection of low precipitation rates (<0.5 mm/h) and falling snow.
A few preliminary assessments of GPM over India and China (Prakash et al., 2016a,
2016b; Tang et al., 2016a) suggest an improvement over TMPA. For 2014 monsoon (Prakash
et al., 2016b) reported that Integrated Multi-satellitE Retrievals for GPM (IMERG), which is
a level three multi-satellite precipitation algorithm of GPM (Hou et al., 2014), outperformed
TMPA in extreme rainfall detection along the Himalayan foothills in North India and over
North Western India, with slightly reduced false alarms. Tang et al. (2016a) found that
IMERG outperformed TMPA in almost all the indices for every sub-region of mainland
China at 3-hourly and daily temporal resolutions. They also reported that IMERG reproduced
probability density functions more accurately at various precipitation intensities and better



represented the precipitation diurnal cycles. In another work by Prakash et al. (2016a),
IMERG was compared with Global Satellite Mapping of Precipitation (GSMaP) V6 and
TMPA 3B42V7 for the 2014 monsoon over India. It was found that IMERG estimates
represented the mean monsoon rainfall and its variability more realistically, with fewer
missed and false precipitation bias and improvements in the precipitation distribution over
low rainfall rates.
Most of the previous studies that compared satellite and reanalysis precipitation
products for pan-India focused at a grid scale, rather than a basin scale (Prakash et al., 2015,
2016a, 2016b). We focused at a basin scale as it is more relevant in terms of water resources
assessment for policy makers. Also, it provides a clear signal of the utility of the satellite
precipitation products at the required spatial resolution for water managers working at a basin
scale.
In this study, we comprehensively evaluated TRMM 3B42 from 1998-2013 over 86
basins in India and explored systematic biases due to climatology and topography. We then
compared TRMM 3B42 precipitation estimates with IMERG for 2014 and explored if the
systematic biases were reduced in IMERG, and whether IMERG was able to better capture
the low rainfall magnitudes. Finally, we used a macroscale hydrologic model (Variable
Infiltration Capacity (VIC)) to evaluate TRMM and IMERG over a flood prone basin in
Eastern India (Mahanadi River basin) for the year 2014.
**2 Description of the study area, datasets used and methodology**
**2.1 Study area**
The study was conducted over India at a basin scale (Fig. 1a). Water Resources
Information System of India (India-WRIS) divides India into 91 major basins (India, 2014).
In this study, 86 basins were used, with the five excluded basins located in the Jammu and
Kashmir region of Northern India (details included in Supplementary table 1). Also, the
Lakshadweep islands (located off the Indian West coast in the Arabian Sea) and the Andaman
and Nicobar islands (located in the Bay of Bengal) were excluded from the analysis due to
scanty rain-gauge monitoring network.
Most of India experiences a tropical monsoon type of climate receiving an average
annual rainfall of around 1100 mm/year, of which about 70-80% is concentrated during the
monsoon season (June – September). Fig. 1b shows the spatial distribution of rainfall,





calculated using India Meteorological Department (IMD) gridded precipitation dataset
(computed using 31 years (1980-2010) of rainfall time series) over India. The Western Ghats
(located on the Indian West coast) and the North-Eastern basins receive the highest rainfall,
with the magnitude going as high as 3000 mm/year. The Western Ghats receive orographic
rainfall due to the steep topographic gradient that exist from the West to the East, making the
Eastern part of the mountains a leeward area where rainfall is mainly associated with the
passage of lows and depressions developed in the Bay of Bengal (Prakash et al., 2016a).
Details of the orographic features of rainfall over Western Ghats can be found in Tawde and
Singh (2015). The high rainfall in the North-Eastern part of India is associated with
orographic control and multi-scale interactions of monsoon flow (Prakash et al., 2016a).
Basins in the Indo-Gangetic plain and on the East coast receive above average rainfall of
around 1400 mm/year, governed by the tropical monsoons. The hilly tracts of Jammu and
Kashmir situated in North-most part of India receive an annual average rainfall of around
1000 mm/year. The North-west basins, associated with semi-arid type of climate, receive low
annual rainfall ranging from 300-400 mm/year. The basin-wise rainfall is provided in
Supplementary table 1.
Fig. 1c shows the spatial distribution of the basin-wise elevation above mean sea level
(m.s.l). The Northern tract of Jammu and Kashmir comprises the basins with highest
elevations, in between 2500 m to 5000 m above m.s.l. These basins also suffer from scanty
rain monitoring networks, due to which five of these high elevation basins have been ignored
in the analysis (details in Supplementary table 1). High Pitch Mountains are also found in the
North-Eastern basins where basin-wise elevation goes as high as 1400 m above m.s.l. The
Western Ghats are characterized by a very sharp topographic gradient with the elevations
increasing from around 200 m on the West coast to above 600 m above m.s.l as we move
east. This transition results in heavy orographic rainfall on the West coast and leads to the
sharp rainfall contrast on the leeward side of the Western Ghat Mountains. The Indo-
Gangetic plain and the Eastern basins are mostly plateau areas, with basin elevation lying in
between 200-400 m above m.s.l. The semi-arid North-Western basins are also characterized
by plateau land (elevation between 200-300 m above m.s.l). The basin-wise elevation is
provided in Supplementary table 1.
The rainfall-runoff modeling exercise was carried out in the Hirakud catchment of the
Mahanadi River basin (MRB), located on the Eastern coast of India. MRB is one of the
largest Indian basins draining an area of 1,41,000 km$^2$, mostly flowing through the states of





Chattisgarh and Odisha. It is prone to frequent flooding at the downstream, with five major
flood events in the first decade of the 21st century (Jena et al., 2014). On the upstream of the
MRB is a multi-purpose dam (Hirakud) which encompasses catchment area of around 85,200
km$^2$ and spans between 19.5° and 23.8° N latitudes and 80° to 84° E longitudes (Fig. 1d).
Hirakud dam started its operations in 1957 and its upstream does not include any major dam,
although a number of small scale irrigation reservoirs are operational during the monsoon.
The area experiences a tropical monsoon type of climate, with an annual rainfall of around
1500 mm. Agricultural, forest and shrub land account for around 55%, 35% and 7% of the
total basin coverage respectively (Kneis et al., 2014).
**2.2 Datasets used**
IMD gridded rainfall dataset was used as the reference product and Tropical Rainfall
Measuring Mission (TRMM) and Integrated Multi-satellitE Retrievals for GPM (IMERG)
were compared against IMD. A brief summary of the datasets is given in Table 1. A brief
introduction to the three rainfall datasets is given below.
**2.2.1 Gridded IMD and streamflow dataset**
IMD gridded precipitation dataset provides daily rainfall estimates over the Indian
landmass from 1901-2014 at a spatial resolution of 0.25° x 0.25°. It has been developed using
a dense network of rain gauges consisting of 6955 stations and is known to reasonably
capture the heavy orographic rainfall in the Western Ghats, the Northeast and the low rainfall
on the leeward side of the Western Ghats. For a detailed discussion on the evolution of IMD
gridded dataset, refer to Pai et al. (2014).
It is to be noted that IMD measures rainfall accumulation at 8:30 AM Indian Standard
time (IST) or (3:00 AM UTC). The accumulated rainfall for the previous day is provided as
the rainfall estimate for current day. For instance, IMD rainfall estimate at a gauging station
for September 14$^{th}$, 2014 refers to the rainfall accumulation from 8:30 AM IST (3:00 AM
UTC) on September 13$^{th}$, 2014 to 8:30 AM IST (3:00 AM UTC) on September 14$^{th}$, 2014.
Both TRMM and IMERG precipitation estimates were converted to IMD timescale.
The gridded daily minimum and maximum temperature was obtained from IMD at a
spatial resolution of 1° x 1° (Srivastava et al., 2009). Daily wind speed data was obtained
from coupled National Centers for Environmental Prediction (NCEP) and Climate Forecast
System Reanalysis (CFSR) at a spatial resolution of 0.5° x 0.5°. Daily discharge data at the





inflow site of the Hirakud reservoir was obtained from the State Water Resources Department
(Odisha), Hirakud Dam Project, Burla, Sambalpur.
**2.2.2 Tropical Rainfall Measuring Mission (TRMM)**
In order to provide high resolution precipitation dataset in real-time, the TRMM
satellite was launched in late 1997 and it provides 3-hourly rainfall estimates from 1998 to
the current date at a quasi-global coverage (50° N-S) at a spatial resolution of 0.25° x 0.25°
(Huffman et al., 2007). Two variants of TRMM multi-satellite precipitation analysis (TMPA)
are available, a real time product which is available at 3-6 hours latency and the research
product which is available at 2-months latency. TRMM research product makes use of rain
gauge stations from Global Precipitation Climatology Centre (GPCC) to post-process the
TRMM estimates, details of which can be found in Huffman et al. (2007). We used TRMM
research product in this study (henceforth mentioned as TRMM).
**2.2.3 Integrated Multi-SatellitE Retrievals for GPM (IMERG)**
Due to the great success of TMPA mission, Global Precipitation Measurement (GPM)
was launched on February 27, 2014 (Liu, 2016). IMERG is the day-1 multi-satellite
precipitation algorithm for GPM which combines data from TMPA, PERSIANN, CMORPH
and NASA PPS (Precipitation Processing System). For a detailed understanding of the
retrieval algorithm of IMERG, refer to  (Huffman et al., 2014; Liu, 2016).
The major advancement in GPM satellite is the improved sensitivity of sensors
leading to improved detection of low precipitation rates (<0.5 mm/h) and falling snow, a
known shortcoming of TRMM. IMERG is available in 3 variants, (a) Early run (latency ~ 6
hours), (b) Late run (latency ~ 18 hours) and (c) Final run (latency ~ 4 months) (Liu, 2016).
Each product is available at half-hourly temporal and 0.1° x 0.1° spatial resolution. The
spatial coverage is 60° N-S which is planned to be extended to 90° N-S in the near future. We
used the Final run product in our analysis.
**2.3 VIC Hydrological Model**
VIC is a macroscale semi-distributed hydrological model which uses a grid-based
approach to quantify different hydro-meteorological processes by solving water balance and
energy flux equations, specifically designed to represent the surface energy and hydrologic
fluxes at varying scales (Liang et al., 1994, 1996). VIC uses multiple soil layers with variable





infiltration, non-linear baseflow and addresses the sub-grid scale variability in vegetation. A
stand-alone routing model (Lohmann et al., 1996) is used to generate runoff and baseflow at
the outlet of each grid cell, assuming linear and time-invariant runoff transport. The land
surface parameterization (LSP) of VIC is coupled with a routing scheme in which the
drainage system is conceptualized by connected-stem rivers at a grid scale. The routing
model extends the FDTF-ERUHDIT (First Differenced Transfer Function-Excess Rainfall
and Unit Hydrograph by a Deconvolution Iterative Technique) approach (Duband et al.,
1993) with a time scale separation and liberalized Saint-Venant equation type river routing
model. The model assumes runoff transport process to be linear, stable and time invariant.
VIC has been successfully used in a number of global and local hydrologic studies
(Hamlet and Lettenmaier, 1999; Shah and Mishra, 2015; Tong et al., 2014; Wu et al., 2014;
Yong et al., 2012). A recent commentary on the need for process-based evaluation of large-
scale hyper-resolution models by Melsen et al. (2016) provides interesting insights into the
use of VIC at different spatial scales and why we shouldn't just decrease the grid size (hence
increasing the spatial resolution of model) without considering the dominant processes at that
scale. In lines with the discussions in Melsen et al. (2016), VIC was run at a grid size of 0.5°
x 0.5°.

**2.4 Methodology**

All the analysis was performed at a basin scale. Basin-wise mean areal rainfall was
calculated for all the three rainfall products (IMD, TRMM and IMERG) using Thiessen
Polygon method for their respective periods of availability.
In order to statistically evaluate the precipitation products, two skill measures were
used (Pearson correlation (R) and percentage bias (Pbias/bias)) along with two threshold
statistics (probability of detection (POD) and false alarm ratio (FAR)). Table 2 shows the
contingency table and Table 3 provides a summary of the statistical indices.
All the statistical inferences were drawn for the overall time series, and then
separately for the different rainfall regimes. Table 4 shows the criterion to segregate the
rainfall time series into different components. For computing POD and FAR for different
rainfall regime, a threshold is required. The 25th percentile value was selected as the
threshold for low rainfall regime, 50th percentile for medium regime, 75th percentile for high

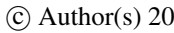



248 rainfall regime and 95th percentile for very high rainfall regime. The statistical indices were

249 calculated basin-wise.

250  In order to identify systematic bias in the satellite products, one meteorologic index

251 (long term basin mean annual rainfall) and one topographic index (basin mean elevation) was

252 computed for the 86 basins. The long term mean annual rainfall was computed using IMD

253 gridded dataset from 1980 – 2010 (31 years). Basin mean digital elevation model (DEM) was

254 extracted from Shuttle Radar Topography Mission (SRTM) DEM and mean elevation was

255 obtained on a basin-wise scale.

256  Due to the limited availability of IMERG data (starting from 2014), calibration of

257 VIC was done using an approach similar to the one used by Tang et al. (2016b). First, VIC

258 was calibrated (2000-2011) and validated (2011-2014) using gridded IMD precipitation time

259 series. VIC was then calibrated (2000-2011) and validated (2011-2014) with TRMM

260 precipitation time series. Further, both the IMD and TRMM calibrated models were validated

261 with IMERG and TRMM for the year 2014 (from April 1, 2014 to December 31st, 2014).

262 The year 2000 was used as a warm up period for the model.

263  In line with the recent discussion by McCuen (2016) on the correct usage of statistical

264 and graphical indices to evaluate model calibration and validation, four statistical parameters

265 (Nash Sutcliffe efficiency (NSE), Percentage bias (Pbias), coefficient of determination ($R^2$)

266 along with its significance probability (p-value) and root mean squared error (RMSE)) were

267 used to evaluate the runoff simulations from VIC. Table 3 provides a summary of these

268 indices.

269 **3 Results**

270  All the TRMM statistics were obtained for two distinct periods (1998-2013 and

271 2014). For the year 2014, the IMERG precipitation estimates were available from March 12,

272 2014. Therefore, the TRMM statistics for the year 2014 were obtained from March 12, 2014

273 to December 31, 2014. Henceforth, for the sake of convenience, statistics of TRMM-R refers

274 to the time period 1998-2013, statistics of TRMM and IMERG refers to the time period

275 March 12, 2014 to December 31, 2014.

276 **3.1 Scatterplots**



277       Fig. 2.1 shows the scatterplot of IMERG and TRMM with respect to IMD

precipitation combining data from all the 86 basins for the year 2014. Both IMERG and
TRMM show quite similar skills with correlation values above 0.8, with IMERG showing
better correlation in 60 out of 86 basins. On looking at the scatterplots for individual basins
(Fig. 2.2), IMERG tends to be better correlated to IMD than TRMM. It can be seen that the
correlation values go as high as 0.96 for IMERG (and 0.94 for TRMM) with a very uniform
spread across the 1:1 line for the five best basins (Figs. 2.2a–e) (decided on the basis of
correlation of IMERG with IMD in 2014). These basins are situated in the flat Deccan
Plateau belt in South-central India (mostly concentrated in Tapi and Godavari basins). For
the other five basins (Figs. 2.2f–j), the poor correlation is due to the gross overestimation of
IMERG/TRMM over IMD. Four of these five basins are situated in the high elevation basins
in Northern India, which hints at a systematic dependence of IMERG/TRMM estimates with
elevation. This is explored in detail in section e.
**3.2 Basin-wise correlation**

291       Basin-wise correlation was computed for retrospective analysis of TRMM-R and to

compare TRMM and IMERG rainfall estimates for the year 2014. Fig. 3 suggests that
IMERG gives slightly better rainfall estimate than TRMM for all rainfall regimes (with
IMERG showing higher correlation for the year 2014 for 60, 52, 52 and 55 out of 86 basins
for overall, low, medium and high rainfall regimes). IMERG shows a correlation coefficient
higher than 0.8 (for overall time series) for 73 out of 86 basins, compared to 68 basins for
TRMM and higher than 0.9 for 20 basins compared to 13 for TRMM. The decomposition of
the overall time series into different rainfall regime reduces the correlation, which can be
attributed to temporal smoothening in longer time series.

300       The spatial maps (Fig. 4) provide an illustration of the slight improvement of IMERG

over TRMM with spatially coherent patterns. In general, both TRMM and IMERG show high
basin-wise correlation values for the overall time series. In the overall spatial maps (Figs. 4b–
c), for the year 2014, TRMM and IMERG show similar skill, with IMERG capturing the
rainfall slightly better in Central and Southern India. Both show similar skill in the high
rainfall areas of the Western Ghats and the North Eastern basins. IMERG gives slightly better
estimates in the high elevation basins in North India. There is no significant improvement in
the basins located on the Eastern coast (like the Mahanadi river basin). TRMM provides
slightly better estimates of rainfall in the semi-arid basins located in the North Western states



of India (Rajasthan). It is to be noted that TRMM statistics for 2014 are much better than its retrospective statistics (TRMM-R) with spatial coherent trends.

The low rainfall estimates (Figs. 4d–f) over the semi-arid North Western basins are slightly better for TRMM compared to IMERG. IMERG captures low rainfall better over the Indo-Gangetic plain. Both IMERG and TRMM show similar trends over the Western Ghats, North-Eastern basins, Eastern coast and over the Deccan Plateau. IMERG doesn't capture the low rainfall regime over the Upper Indus basin (in Northern India) and over the upper Bhima and the upper Godavari basin (in the Deccan plateau belt).

The medium rainfall estimates (Figs. 4g–i) are best represented in Central India and over the Deccan Plateau by TRMM and IMERG. Both show similar statistics over the Western Ghats and basins in North-Eastern and Eastern coast of India. TRMM slightly outperforms IMERG in the North-Western basin of Rajasthan, a trend also found in the low rainfall regime. IMERG doesn't capture the medium rainfall trends over the Upper Indus basin (in Northern India). In general, TRMM-R medium rainfall estimates are best correlated in the semi-arid region of Rajasthan (North-Western basins) and in Central India. There is not much variability in the correlation of medium rainfall trends of TRMM-R, with correlation coefficient mostly around 0.5 for entire India, except for the high elevation Upper Indus basin.

The high rainfall estimates (Figs. 4j–k) show highest correlation in the Deccan Plateau belt, higher elevation basins in Northern India, the Western Ghats and the East coast basins (except for the Southern-most basin) for TRMM and IMERG. High rainfall estimates of TRMM are better correlated than IMERG in the North-Eastern basins of Brahmaputra and Barak and the North-Western basins of Rajasthan. Both show similar correlation over the high elevation basins in the North and over the Western Ghats. IMERG outperforms TRMM in the rain-shadow area of the Western Ghats and in the South-Eastern basins of Pennar and Cauvery. Retrospective maps of TRMM-R (Fig. 4j) suggest that high rainfall is adequately captured in the Indo-Gangetic plain, Western Ghats, North-Western basins of Rajasthan, South-Eastern basins of Pennar and Cauvery and the Eastern coast basins of Central India. However, TRMM gives very low correlation values for the rain-shadow belt of the Western Ghats, suggesting that it doesn't capture the steep orographic gradient. The high rainfall estimates of TRMM-R give modest correlation in the North-Eastern basins, high elevation basins in Northern India and the West most basins of the South (Varrar and Periyar).



### 3.3 Basin-wise bias


Basin-wise bias was computed for retrospective analysis of TRMM-R and to compare
TRMM and IMERG rainfall estimates for the year 2014. Although, IMERG tends to give
slightly better correlation on a basin-wise scale (Fig. 3a), Fig. 5a suggests that it also
enhances the bias in the product. The bias plot for the low rainfall regime (Fig. 5b) suggests
that TRMM is more negatively biased than IMERG for 75 out of 86 basins. Negative bias
indicates overestimation, which is a known problem with TRMM as its sensors cannot detect
very low rainfall magnitudes (<0.5 mm/hour) (Hou et al., 2014). If it detects a low intensity
storm, it is most likely to overestimate it which can be clearly seen in Fig. 5b. IMERG tends
to give a better estimate of low rainfall magnitudes with smaller negative biases for 75 out of
86 basins, due to the sensor improvements in the GPM mission (Huffman et al., 2014). For
the medium rainfall magnitudes, IMERG slightly increased the bias in the majority of basins
(63 out of 86). In TRMM, there were 18 basins which showed positive bias which was
increased to 38 in IMERG. However, this is not to be misunderstood as a decay in skill as in
TRMM there were 28 basins which were relatively unbiased (-10% <=bias <= 10%) which
was increased to 37 in IMERG. IMERG tends to increase the variability of bias in the high
rainfall regime (Fig. 5d). For the high rainfall estimates, TRMM has 57 basins whose bias lies
between -20% to +20% which is decreased to 52 in IMERG. In TRMM, 57 basins showed
positive bias (implying underprediction) which was reduced to 48 basins in IMERG. This
suggests a reduction in systematic underprediction, although with greater variability in bias in
IMERG for the high rainfall regime.
The spatial maps for the overall rainfall time series (Figs. 6a-c) suggests similar bias
patterns in TRMM and IMERG with spatial coherent trends throughout most of India.
IMERG gives slightly lower bias over the high elevation basins of North India (Upper Indus
basin) and slightly higher bias over the North Eastern basins (of Brahmaputra and Barak) and
the West flowing rivers of Kutch on the Western coast in the state of Gujarat. IMERG gives a
large negative bias (overprediction) over Upper and Middle Godavari basin (in Deccan
Plateau belt) which suggests that the sharp topographic gradient is not well captured.
Retrospective maps of TRMM-R suggest an underestimation over high elevation basins in
Northern India (Indus, Jhelum and Chenab basins). However, TRMM captures the heavy
precipitation on the Western Ghats well with very low biases.


The low rainfall spatial maps (Figs. 6d–f) show the large overprediction (negative
bias) by TRMM (1998-2013 and 2014) which is improved in IMERG. The improvement is
most prominent in the North Eastern basins (of Brahmaputra and Barak), Central India
(Mahi, Chambal and the Indo-Gangetic plain), rain-shadow area of the Western Ghats and the
South-Eastern coast. IMERG shows gross overprediction over Luni basin (near the Western
coast of Rajasthan). Retrospective TRMM-R maps for low rainfall regime (Fig. 6d) show that
the low rainfall was best captured in high rainfall areas of the Western Ghats, the Indo-
Gangetic plain and the Eastern coastal basins, which is not very surprising as TRMM doesn't
detect low rainfall magnitudes very well, thus suffering from overprediction in arid and semi-
arid basins. Improvement in the low rainfall sensors in IMERG has improved low rainfall
estimates, but it still suffers from gross overprediction in semi-arid areas (as evident in the
semi-arid basins in North-West India (Fig. 6f).
The medium rainfall spatial maps (Figs. 6g–i) suggest very similar spatial bias pattern
in TRMM and IMERG, with low biases in most of the basins. Both TRMM and IMERG
suffer from underprediction (positive bias) in the high elevation Northern basins (of Indus
and Jhelum), although IMERG seem to be less biased than TRMM. Both show similar trends
in the Western Ghats, with very low bias. However, both the products show large negative
bias (overprediction) in the Middle Godavari basin, unable to capture the sharp topographic
gradient in the region. IMERG slightly overpredicts rainfall in the North Eastern basins (of
Brahmaputra and Barak). The retrospective TRMM maps for medium rainfall (Fig. 6g) show
almost constant bias (almost unbiased) over entire India, except over the Western Ghats
(slightly positive bias (slight underprediction)) and high elevation Northern basins of Indus
and Jhelum (positive bias (strong underprediction)).
The high rainfall spatial maps (Figs. 6j–l) suggest similar spatial pattern in TRMM
and IMERG, with slight negative bias over majority of the basins. The high rainfall in the
Western Ghats is well represented in TRMM and IMERG, with overprediction in the leeward
side of the Western Ghats, suggesting that IMERG is unable to capture the sharp topographic
gradients. IMERG shows slightly greater bias (implying greater underprediction) in the high
rainfall areas of the North Eastern basins. IMERG gives a better estimate (still underpredicts)
in the high elevation basins in Northern India. Both IMERG and TRMM give similar bias
pattern in the Indo-Gangetic plain and the semi-arid areas of the North-West. The
retrospective TRMM-R map for high rainfall (Fig. 6j) suggests that TRMM slightly
overpredicts high rainfall in majority of India (Indo-Gangetic plain, Deccan Plateau, rain-



shadow area of the Western Ghats). However, it suffers from gross underestimation in the
high elevation basins of Northern India (Indus, Jhelum and Chenab). It is clearly observed
that the high elevation basins are an outlier in most of the analysis, a systematic dependence
of bias with elevation may be an underlying trend which is further explored in section *e*.
**3.4 Threshold statistics**
Basin-wise POD and FAR was computed for retrospective analysis of TRMM-R and
for the comparison of TRMM with IMERG (Figs. 7 and 8). Four rainfall thresholds were
chosen, representative of different rainfall regimes (low threshold: 25 percentile, medium
threshold: 50 percentile, high threshold: 75 percentile and very high threshold: 95 percentile).
Increasing rainfall threshold leads to deteriorating trends in POD and FAR across majority of
the basins, with decreasing POD and increasing FAR.
For the low rainfall threshold, IMERG gives higher POD than TRMM for 62 basins,
with the major improvement in the Western region of Gujarat (Luni, Bhadar and Setrunji
basins) (Figs. 7b,c). There is less spatial variability in POD for both TRMM and IMERG at
low rainfall threshold with POD above 0.9 for 75 basins for IMERG and 63 basins for
TRMM. The average POD (low rainfall threshold) across basins is 0.95 for IMERG and 0.91
for TRMM. For the medium rainfall threshold, IMERG outperforms TRMM in 39 basins
with TRMM giving a higher POD in 37 basins; both the products give similar POD in 10
basins. The average POD (medium rainfall threshold) across basins is 0.87 for both IMERG
and TRMM. Notably, IMERG gives lower POD (medium rainfall threshold) in 2 (Barak and
Brahmaputra lower sub-basin) out of the 3 North-Eastern basins, and higher POD (medium
rainfall threshold) in the semi-arid basins of Rajasthan and Gujarat (Luni, Bhadar and
Setrunji basins) (Figs. 7e,f). For the high rainfall threshold, TRMM outperforms IMERG in
45 basins with IMERG giving a higher POD in 32 basins, both the products give similar POD
in 9 basins. The average POD (high rainfall threshold) across basins is 0.76 for IMERG and
0.77 for TRMM. There is notable fall in performance in all the 3 North-Western basins.
IMERG gives slightly higher POD (high rainfall threshold) in the high elevation Northern
basins (Upper Indus and Jhelum basins) (Figs. 7h,i). For the very high rainfall threshold,
IMERG outperforms TRMM in 44 basins with TRMM giving a higher POD in 27 basins;
both the products give similar POD in 15 basins. The average POD (very high rainfall
threshold) across basins is 0.72 for IMERG and 0.7 for TRMM. At very high rainfall
threshold, it's clear that POD of IMERG is worse for all the 3 North-Eastern basins and over





the semi-arid basins of Rajasthan and Gujarat (Figs. 7k,i). There is slight improvement in
POD values for the high elevation Northern basins (Chenab, Ravi, Beas and Satulaj basins).

439        At low rainfall threshold, TRMM gives higher FAR than IMERG in 42 basins with

IMERG giving a higher FAR in 40 basins; both the products give similar FAR in 4 basins.
The average FAR (low rainfall threshold) across basins is 0.24 for TRMM and 0.22 for
IMERG. For the medium rainfall threshold, IMERG outperforms TRMM (with lower FAR)
in 53 basins with TRMM giving lower FAR in 26 basins; both the products give similar FAR
in 7 basins. The average FAR (medium rainfall threshold) across basins is 0.22 for TRMM
and 0.19 for IMERG. Notably, IMERG outperforms TRMM at low and medium rainfall
thresholds giving lower FAR in the Western basins of Gujarat (Luni and Setrunji basins)
(Figs. 8b,c,e,f). For the high rainfall threshold, IMERG outperforms TRMM in 67 basins
(lower FAR) with TRMM giving a lower FAR in 15 basins; both the products give similar
FAR in 4 basins. The average FAR (high rainfall threshold) across basins is 0.18 for IMERG
and 0.22 for TRMM. Slightly reduced FAR are seen in Central India (Yamuna and Chambal
basins) and the North-Eastern basins (Brahmaputra basin) in IMERG at high rainfall
threshold (Figs. 8h,i).For the very high rainfall threshold, IMERG outperforms TRMM in 64
basins (lower FAR) with TRMM giving a lower FAR in 17 basins; both the products give
similar FAR in 5 basins. The average FAR (very high rainfall threshold) across basins is 0.33
for IMERG and 0.41 for TRMM. There are notably fewer false alarms in IMERG estimates
over the Northern, North-Eastern basins and the Western Ghats at very high thresholds. Both
products give similar FAR (very high threshold) along the Eastern coast and Deccan Plateau
basins.

459        POD for TRMM-R suggests decreasing POD and increasing FAR with increasing

rainfall threshold (Figs. 7a,d,g,j, Figs. 8a,d,g,j). The average POD across basins is 0.89, 0.85,
0.77 and 0.66 for low, medium, high and very high rainfall thresholds, respectively. The
respective FAR values are 0.26, 0.22, 0.21 and 0.43. At high and very high threshold, POD
drops significantly over the high elevation Northern basins and high rainfall North-Eastern
basins and the Western Ghats) (Figs. 7g,j). High FAR is recorded in the basins in Gujarat
(Luni and Setrunji) and Central India (Bhadar and Chambal) at low and medium rainfall
threshold (Figs. 8a,d) suggesting TRMM creates a lot of false alarms at low and medium
rainfall magnitudes. There is a sharp contrast between FAR at high and very high thresholds,
with low FAR at high rainfall threshold (75 percentile) and high FAR at very high threshold
(95 percentile) (Figs. 8g,j). This suggests that TRMM-R creates a lot of false alarms at very



high rainfall thresholds, especially in the North-Eastern, Northern and extreme Southern
basins (Fig 8j).

**3.5 Systematic error in satellite estimates as a function of annual rainfall and mean**
**elevation**

The satellite precipitation estimates were evaluated against a climatologic parameter
(long term annual rainfall of basin) and a topographic parameter (basin mean elevation). Fig.
9 describes the relationship between mean annual precipitation and mean elevation by
considering the point values for 86 basins. It was found that there is no systematic
dependence between the climatologic and topographic parameter (R = 0.07) and they can be
considered as independent (implying minimal interference).
TRMM-R rainfall estimates exhibited strong systematic dependence of bias and
correlation with basin wise mean rainfall at low and medium rainfall estimates (Figs. 10 and
11). At low rainfall regime, TRMM-R estimates for basins experiencing low annual rainfall
were found to be strongly negatively biased (Fig. 10b), implying significant overprediction.
The bias values improved drastically for basins experiencing higher annual rainfall. This is
also reflected in the correlation plots (Fig. 11b), where a positive correlation between basin-
wise correlation and annual rainfall (R = 0.3) implies improved estimates of low rainfall at
basins which experience high annual rainfall. At the medium rainfall regime, TRMM-R
estimates showed higher bias (implying underprediction) and lower correlation (reduced
skill) in basins receiving higher annual rainfall, with a sharp drop in correlation for heavy
rainfall basins (Figs. 10c and 11c). At high rainfall regime, the systematic bias was reduced,
both in terms of percent bias and correlation, implying that there is no significant difference
in TRMM-R estimates of high rainfall, in basins receiving low/high annual rainfall.
For the year 2014, both IMERG and TRMM showed increasing bias as a function of
increasing annual rainfall for all the rainfall regimes (Fig. 12), with the systematic
dependence strongly reduced in IMERG estimates for the medium rainfall regime. For the
low rainfall regime, bias and correlation values improve for basins receiving higher rainfall
(Figs. 12b and 13b). TRMM and IMERG showed similar systematic dependence on annual
rainfall at low rainfall regime, with correlation values between basin wise correlation and
annual rainfall equal to 0.38 and 0.39 for TRMM and IMERG, respectively. For the medium
rainfall regime, both IMERG and TRMM showed increasing bias with increasing annual
basin-wise rainfall (Fig. 12c). However, there was a strong reduction in the systematic bias



component in IMERG, with correlation between basin-wise bias and rainfall decreasing from
0.43 (for TRMM) to 0.3 (for IMERG). At medium rainfall, a substantial skill was lost in
terms of decreasing correlation for basins receiving high rainfall (Fig. 13c). This systematic
dependence wasn't reduced in IMERG estimates, with correlation values between basin-wise
correlation and rainfall as -0.45 for TRMM and -0.44 for IMERG. At high rainfall regime,
bias was higher for basins which received more rainfall, implying greater underprediction in
basins with heavy rainfall magnitude (Fig. 12d). This systematic bias wasn't reduced in
IMERG estimates. No systematic dependence was found in the correlation of
IMERG/TRMM estimates with basin-wise rainfall (Fig. 13d).
TRMM-R rainfall estimates exhibited very strong dependence on mean basin
elevation, with decreasing skill (higher bias and lower correlation) in basins with high mean
elevation (Figs. 14 and 15). For the low rainfall regime, a correlation coefficient (between
basin-wise bias and elevation) of (-0.08) (Fig. 14b) may suggest that there is no systematic
dependence between elevation and bias. For medium and high rainfall regimes (Figs. 14c, d),
bias values increase drastically for high elevation basins (especially for basins with mean
elevation > 2000 m), implying underprediction at higher elevations. The corresponding
correlation values (Figs. 15c, d) also suggest reduced skill at higher elevation basins.
For the year 2014, except at low rainfall magnitude, bias increases with mean basin
elevation for TRMM and IMERG rainfall estimates (Fig. 16). This systematic dependence of
bias on basin elevation is improved in IMERG estimates, with the correlation between basin-
wise bias and elevation reducing from 0.43 to 0.32 for medium rainfall regime (Fig. 16c) and
from 0.31 to 0.08 for high rainfall regime (Fig. 16d). It's interesting to note that the same is
not seen for the correlation plots (Fig. 17). For the low rainfall regime (Fig. 17b), IMERG
estimates exhibit stronger systematic relationship between basin-wise correlation and
elevation, with strongly decreasing correlation with elevation than TRMM. At medium
rainfall intensity (Fig. 17c), both TRMM and IMERG show decreasing skill with increasing
elevation. This systematic dependence is again stronger in IMERG than TRMM, as reflected
in the higher negative correlation between basin-wise correlation and elevation in medium
rainfall IMERG estimates (Fig. 17c). For the high rainfall intensity (Fig. 17d), both IMERG
and TRMM do not show any systematic dependence of skill with elevation.
**3.6 Rainfall-runoff modeling**





Rainfall-runoff modeling was carried out over Hirakud catchment of Mahanadi River
basin with the calibration and validation periods as 2000-2011 and 2012-2014, respectively.
VIC was first calibrated with IMD gridded precipitation and then with TRMM3B42 V7. The
two calibrated models were then forced with TRMM and IMERG precipitation forcing for
the year 2014 (April – December). Table 5 shows the model performance.
VIC was successfully calibrated using IMD (NSE = 0.83 for calibration and 0.86 for
validation) and TRMM (NSE = 0.72 for calibration and 0.73 for validation). The IMD
calibrated model showed better simulations compared to the TRMM calibrated model, with
higher NSE, coefficient of determination and lower bias and RMSE. TRMM calibrated model
showed slight overprediction (negative bias) (Table 5).
The IMERG simulations with IMD and TRMM calibrated models were slightly
inferior in comparison with TRMM simulations for 2014 (Table 5, Fig. 18). The IMERG
simulations with TRMM calibrated model reported higher NSE and coefficient of
determination, with lower bias and RMSE, which might be due to the fact that TRMM and
IMERG are both satellite products and exhibit similar spatio-temporal trends. The high
negative bias in IMERG simulations (with IMD and TRM calibrated models) showed
significant overprediction compared to TRMM.
Both TRMM and IMERG underestimated the magnitude of the two major peaks (flow
> 15000 m$^3$/s) in 2014. However, the phase was well captured by both IMERG and TRMM.
Apart from the two major peaks, IMERG overestimated flow for the majority of the time in
both IMD and TRMM calibrated VIC model (hence the negative bias value), and thus was
inferior in performance to TRMM.  This suggests that the use of an appropriate post-
processor (in form of real-time error updation) could tremendously benefit the flow
simulations, which might be an interesting study for the future.
**4 Conclusions**
TRMM 3B42 and IMERG precipitation estimates were comprehensively evaluated over
86 basins in India. TRMM 3B42 was analysed for two distinct time periods, the retrospective
analysis was carried out from 1998-2013 and the current estimates were compared with
IMERG for the year 2014 (March 12[th] 2014 – December 31[st] 2014). The systematic biases in
both the estimates were explored with respect to a climatologic parameter (basin mean annual
rainfall) and a topographic parameter (basin mean elevation). Finally, TRMM and IMERG





were hydrologically evaluated by carrying out rainfall-runoff modeling over Hirakud catchment of Mahanadi River basin, a flood prone basin in Eastern India. The results of the study are summarized as:

1. IMERG rainfall estimates were found to be better than TRMM at all rainfall intensities. IMERG outperformed TRMM in 60, 52, 52 and 55 out of 86 basins for overall, low, medium and high rainfall regimes.

2. IMERG gave better estimates of low rainfall magnitudes with smaller negative biases in 75 out of the 86 basins analysed, which suggests that the sensor improvement in IMERG satellite translated into better low rainfall estimation. IMERG captured the low rainfall magnitudes better over the Indo-Gangetic plain, North Eastern basins of Brahmaputra and Barak, Central India (Mahi, Chambal and the Indo-Gangetic plain) and the rain shadow area of the Western Ghats. However, for the semi-arid North Western basins, TRMM low rainfall estimates outperformed IMERG.

3. The high rainfall estimates of IMERG outperformed TRMM in the rain-shadow area of the Western Ghats, the high elevation basins of the North and the South-Eastern basins of Pennar and Cauvery. However, TRMM did a better job in the North-Eastern basins of Brahmaputra and Barak and the North-Western basins of Rajasthan. Interestingly, IMERG reduced the systematic underprediction over TRMM although with greater variability in bias at high rainfall intensity.

4. Increasing rainfall thresholds lead to deteriorating trends in POD and FAR across majority of basins, with decreasing POD and increasing FAR.

5. The skill of TRMM-R medium rainfall estimates (in terms of Pbias and correlation) was found to exhibit strong systematic dependence on annual rainfall (climatologic parameter), with higher bias and lower correlation in basins which received higher annual rainfall. This systematic dependence was reduced significantly in IMERG estimates. However, no such improvement was found at low and high rainfall intensities.

6. A very strong deteriorating skill (increasing bias and decreasing correlation) was found in TRMM-R rainfall estimates at all intensities in the high elevation basins. This systematic dependence was strongly reduced in IMERG estimates at all rainfall intensities, suggesting IMERG captures the rainfall trends better with respect to topography.

7. Rainfall runoff modeling using VIC model over Hirakud catchment of the Mahanadi River basin gave better results with TRMM as input forcing, rather than IMERG. Both TRMM and IMERG captured the phase of the peak flows, however both underreported





the magnitudes. Low flows were grossly over predicted by IMERG, which led to overall
poor performance with IMERG. As longer timeseries of IMERG is available, it may help
in model performance as IMERG can be used to directly calibrate the model, hence
capturing the fine details in the product.
In essence, IMERG gives reasonable improvement in rainfall estimates across majority of
the Indian basins. However, the improvement was not found to be ground breaking, rather
incremental, suggesting that the GPM mission is a worthy successor of the widely acclaimed
TRMM mission. The most notable improvement in IMERG is the reduction in systematic
error dependence on topography (basin mean elevation), which suggests improvements in the
assimilation of satellite observations. The improved sensitivity of Ku and Ka bands in GPM
satellite resulted in improvement in detection of low rainfall magnitudes. The expected
improvement in IMERG in snow detection could not be verified in this study as India is
mostly a tropical country which receives very less snow. The constant overestimation of low
flow magnitudes in the rainfall-runoff exercise suggest that IMERG may benefit from a post
forecast data assimilation scheme, which is a worthy topic for further research.





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





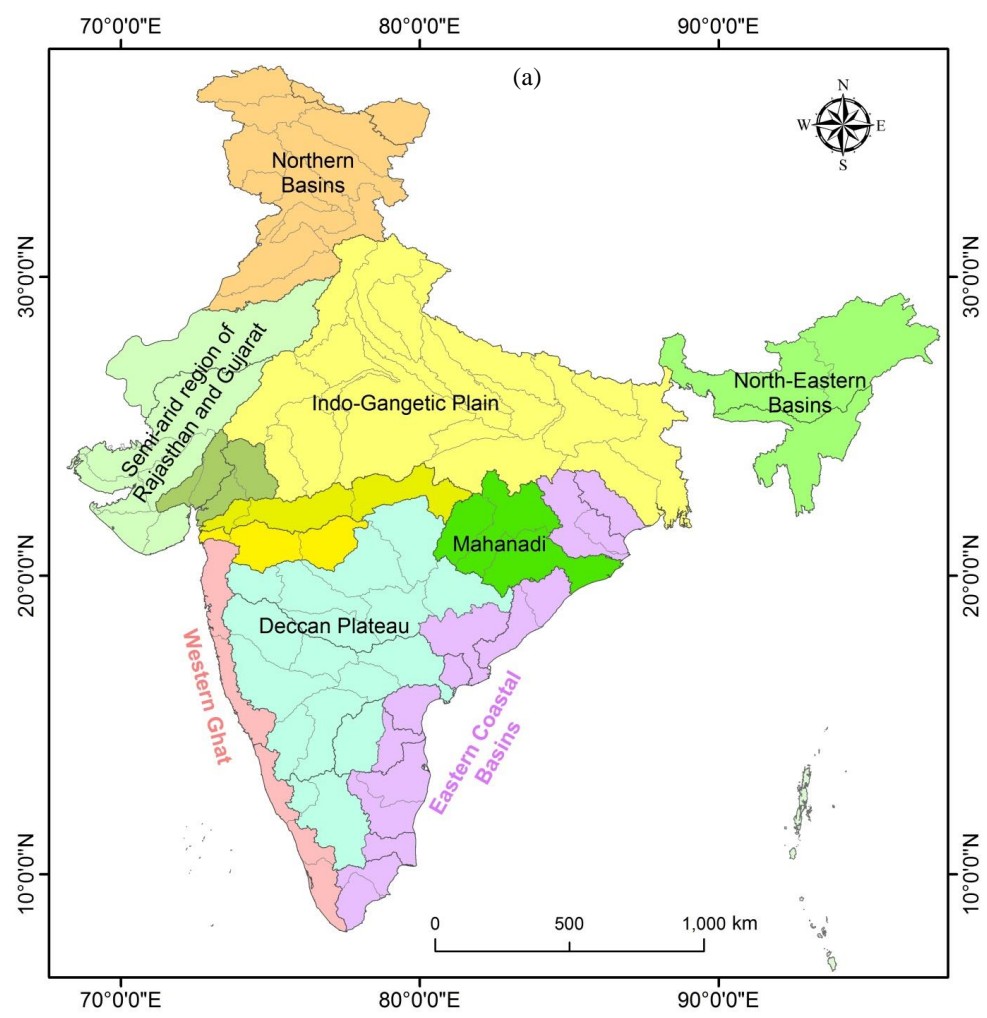


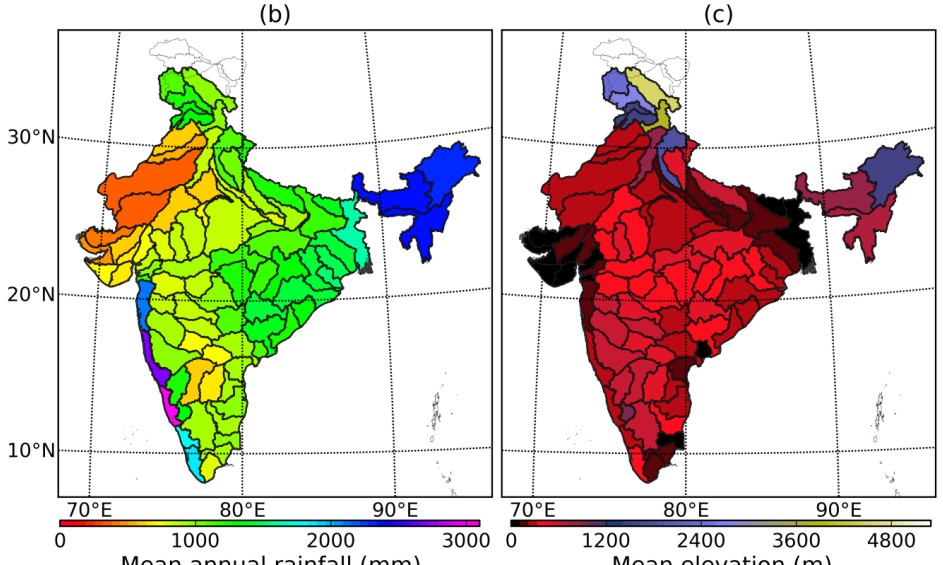



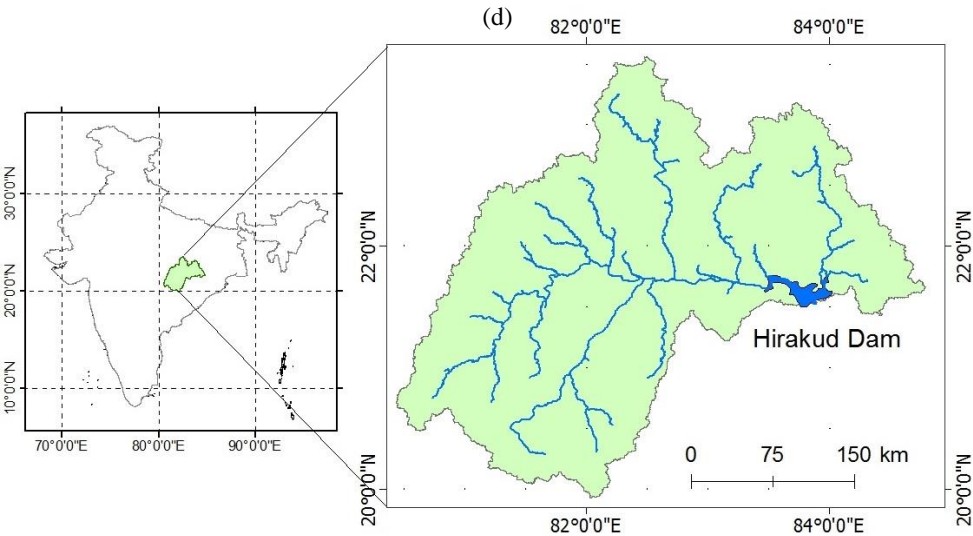


**Figure 1.(a)** Map of the major basins in India, spatial distribution of **(b)** long term average
annual rainfall (calculated from IMD gridded rainfall dataset from years 1980-2010), **(c)**
average elevation above mean sea level (calculated using SRTM DEM) over 86 major basins
in India and (d) map of Hirakud dam catchment of the Mahanadi River basin in Eastern India.





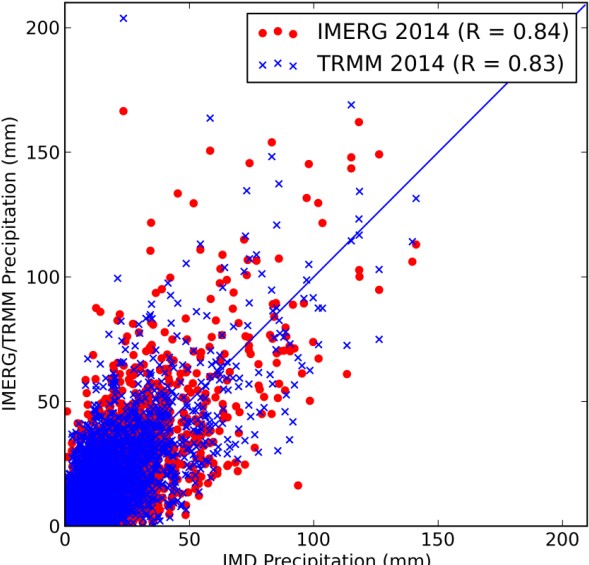


**Figure 2.1** Scatterplot of satellite precipitation products (TRMM and IMERG) vs observed rainfall (IMD) computed over 86 major basins in India (from March 12, 2014 to December 31, 2014).

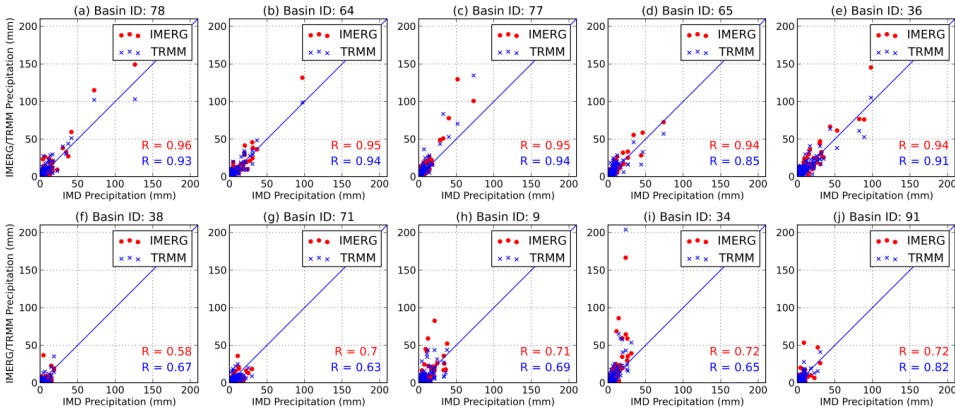


**Figure 2.2.** Scatterplot of satellite precipitation products (TRMM and IMERG) vs observed rainfall (IMD) for **(a)** – **(e)** five best basins in terms of correlation of IMERG with IMD (arranged in descending order) and **(f)** – **(j)** five worse basins in terms of correlation of IMERG with IMD (arranged in ascending order) (from March 12, 2014 to December 31, 2014).





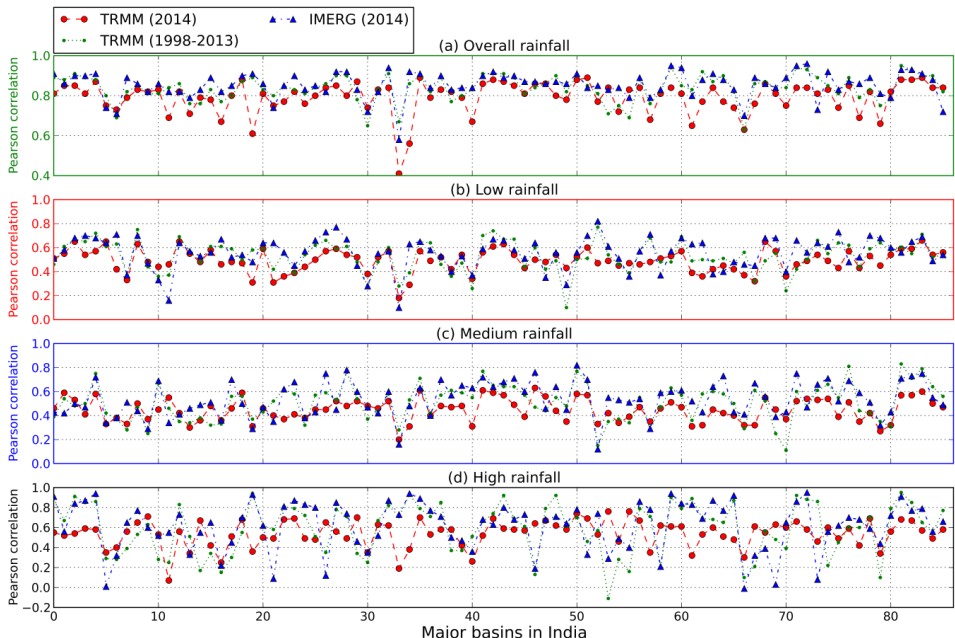

744

**Figure 3.** Correlation of TRMM (1998-2013), TRMM (2014) and IMERG (2014) over 86 major basins in India for **(a)** overall time series and over **(b)** low, **(c)** medium and **(d)** high rainfall regime.





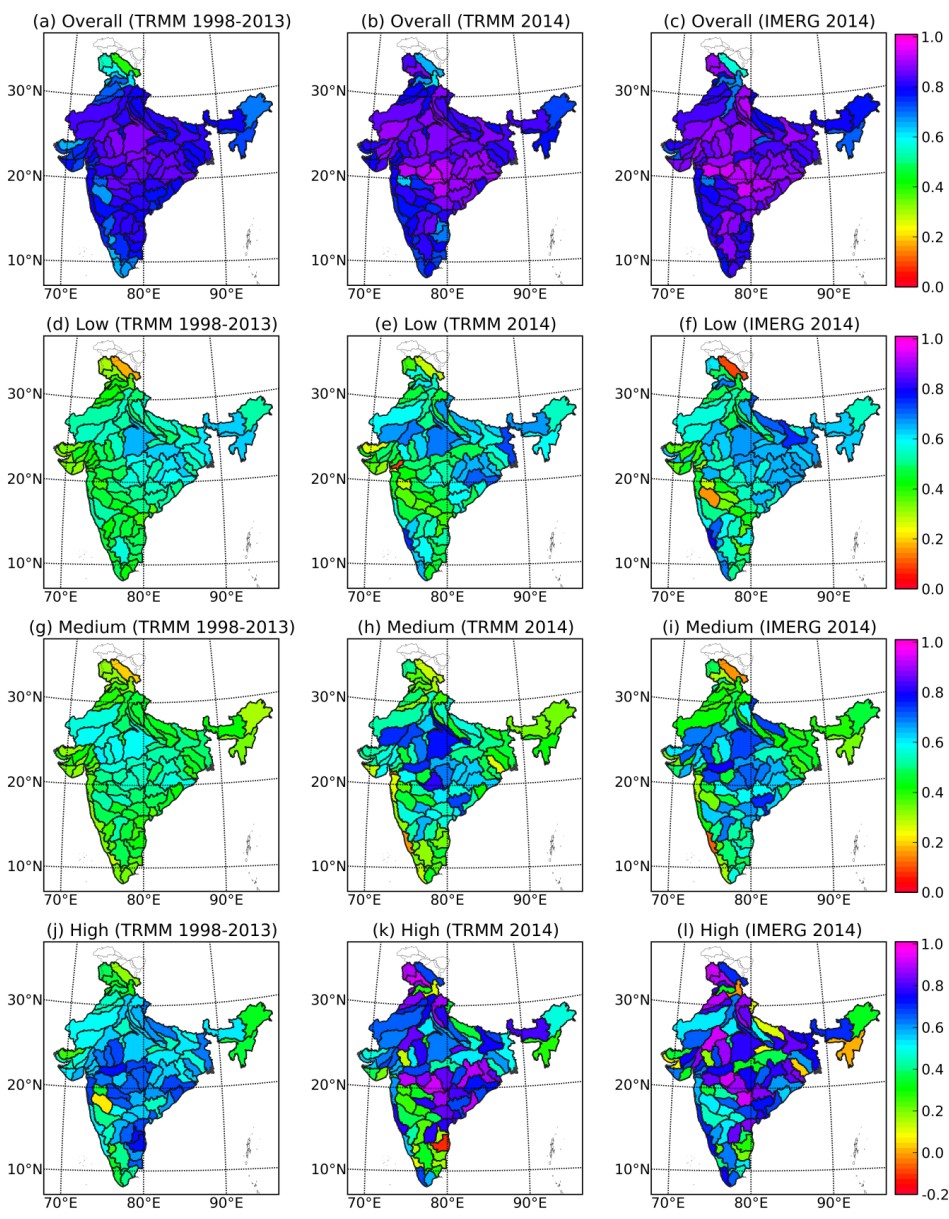

748

**Figure 4.** Spatial representation of correlation of TRMM (1998-2013), TRMM (2014) and
IMERG (2014) over 86 major basins in India for **(a)** – **(c)** overall time series, **(d)** – **(f)** low,
**(g)** – **(i)** medium and **(j)** – **(l)** high rainfall regime.





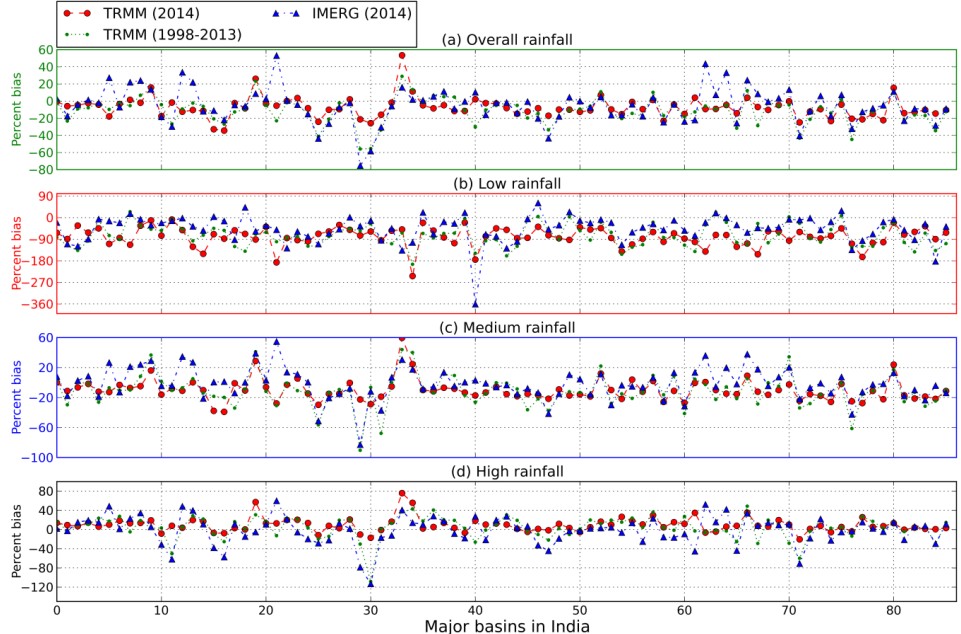

752

**Figure 5.** Percentage bias of TRMM (1998-2013), TRMM (2014) and IMERG (2014) over

86 major basins in India for **(a)** overall time series and over **(b)** low, **(c)** medium and **(d)** high

rainfall regime.





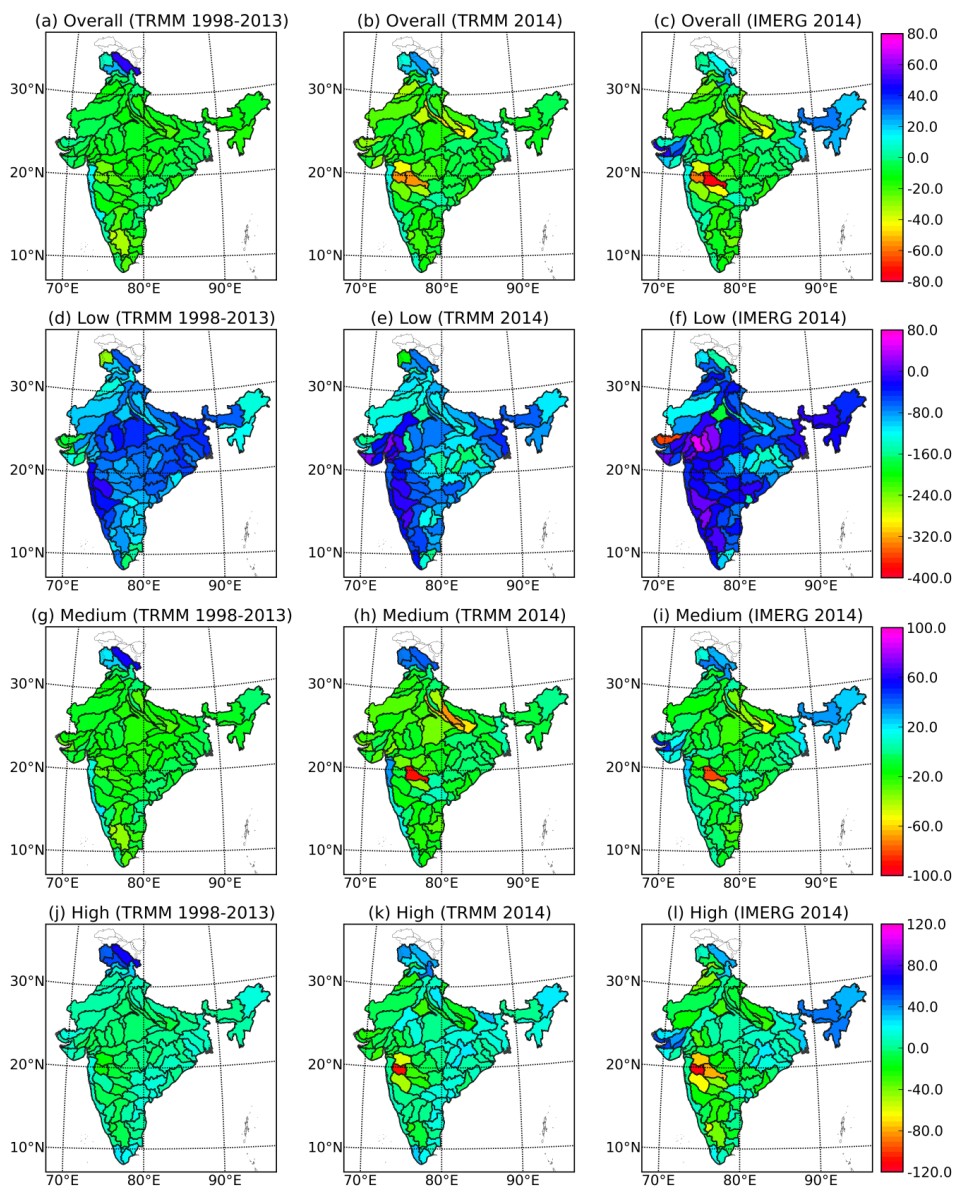

**Figure 6.** Spatial representation of percentage bias of TRMM (1998-2013), TRMM (2014) and IMERG (2014) over 86 major basins in India for **(a)** – **(c)** overall time series and over **(d)** – **(f)** low, **(g)** – **(i)** medium and **(j)** – **(l)** high rainfall regime.





760

**Figure 7.** Spatial representation of probability of detection (POD) for **(a)** – **(c)** low (25 percentile), **(d)** – **(f)** medium (50 percentile), **(g)** – **(i)** high (75 percentile) and **(j)** – **(l)** very high (95 percentile) rainfall threshold for TRMM (1998-2013), TRMM (2014) and IMERG (2014) rainfall estimates over 86 major basins in India.



765

**Figure 8.** Spatial representation of false alarm ratio (FAR) for **(a)** – **(c)** low (25 percentile),

**(d)** – **(f)** medium (50 percentile), **(g)** – **(i)** high (75 percentile) and **(j)** – **(l)** very high (95

percentile) rainfall threshold for TRMM (1998-2013), TRMM (2014) and IMERG (2014)

rainfall estimates over 86 major basins in India.





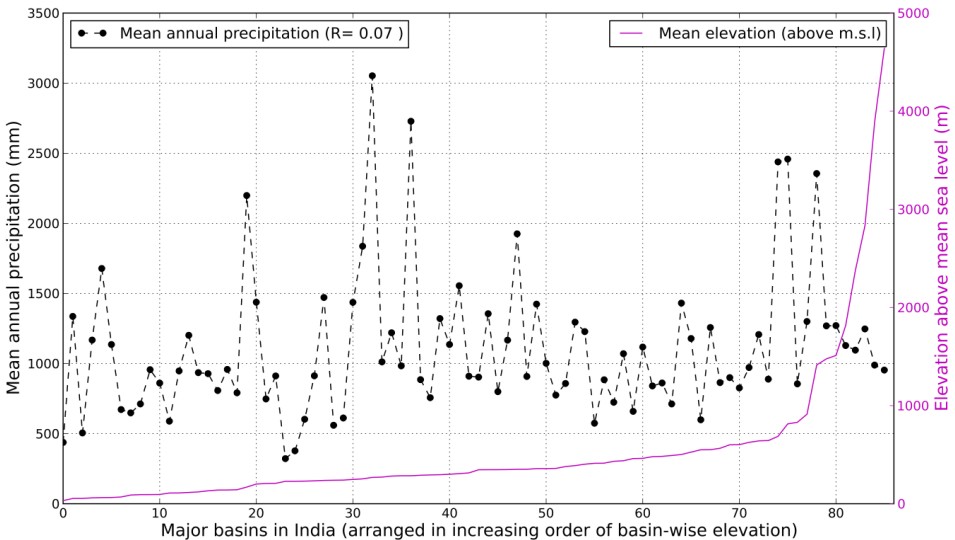

**Figure 9.** Graphical representation of long term average annual rainfall (calculated from IMD gridded rainfall dataset from years 1980-2010) and average elevation above mean sea level for 86 major basins in India (arranged in increasing order of their mean elevation).

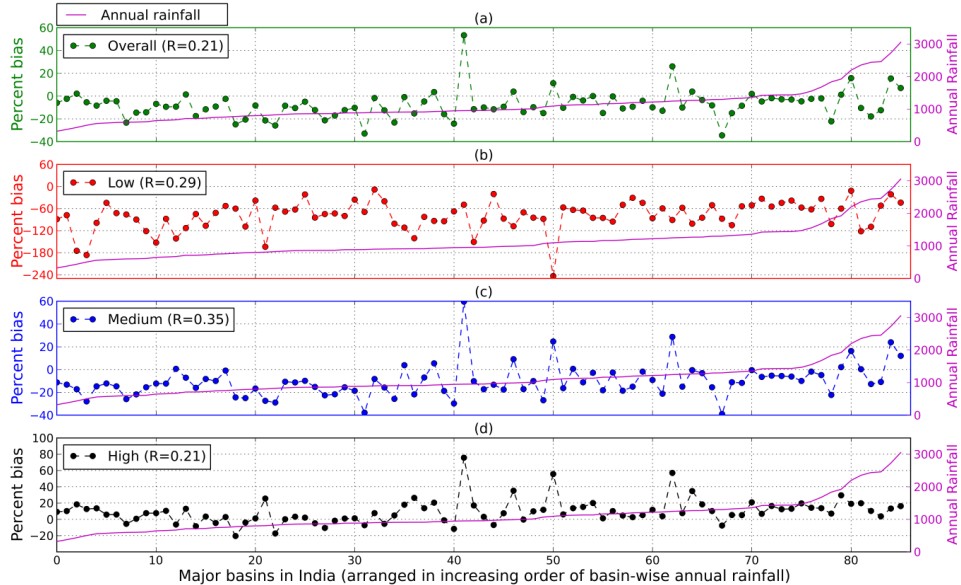


**Figure 10.** Graphical representation of percentage bias of TRMM (1998-2013) arranged in
the increasing order of basin-wise average annual rainfall for **(a)** overall time series and over
**(b)** low, **(c)** medium and **(d)** high rainfall regime for 86 major basins in India.



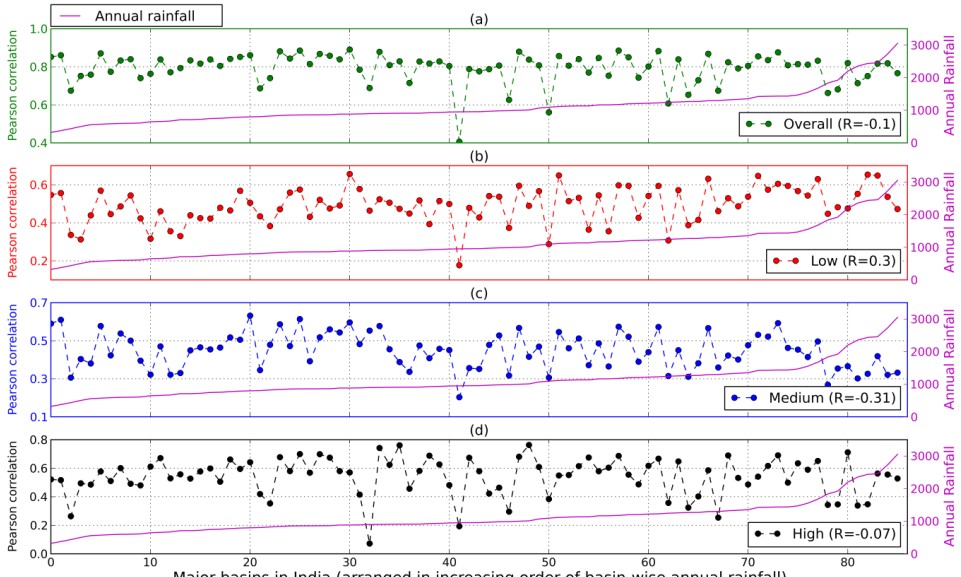

**Figure 11.** Graphical representation of correlation of TRMM (1998-2013) arranged in the increasing order of basin-wise average annual rainfall for **(a)** overall time series and over **(b)** low, **(c)** medium and **(d)** high rainfall regime for 86 major basins in India.

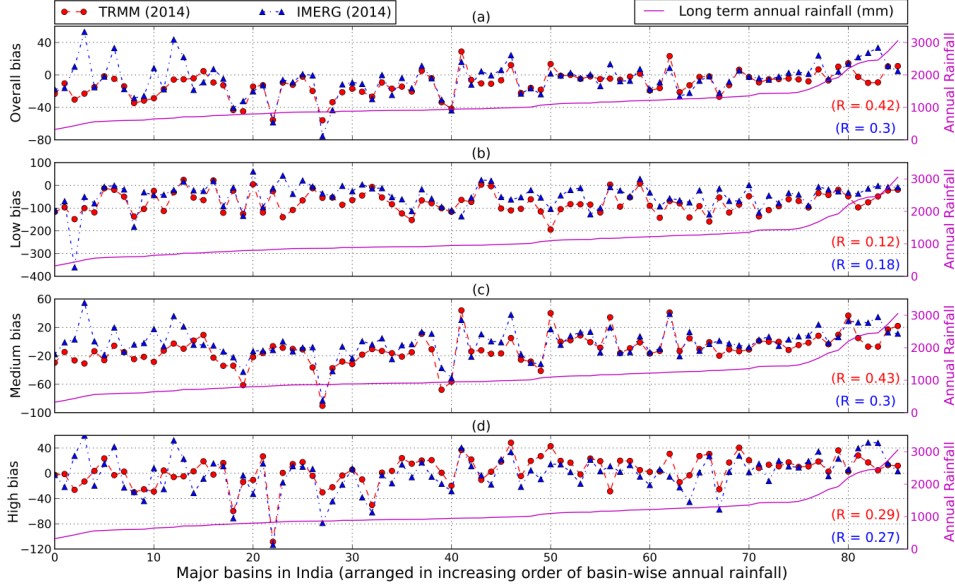

**Figure 12.** Graphical representation of percentage bias of IMERG (2014) and TRMM (2014) arranged in the increasing order of basin-wise average annual rainfall for **(a)** overall time series and over **(b)** low, **(c)** medium and **(d)** high rainfall regime for 86 major basins in India.





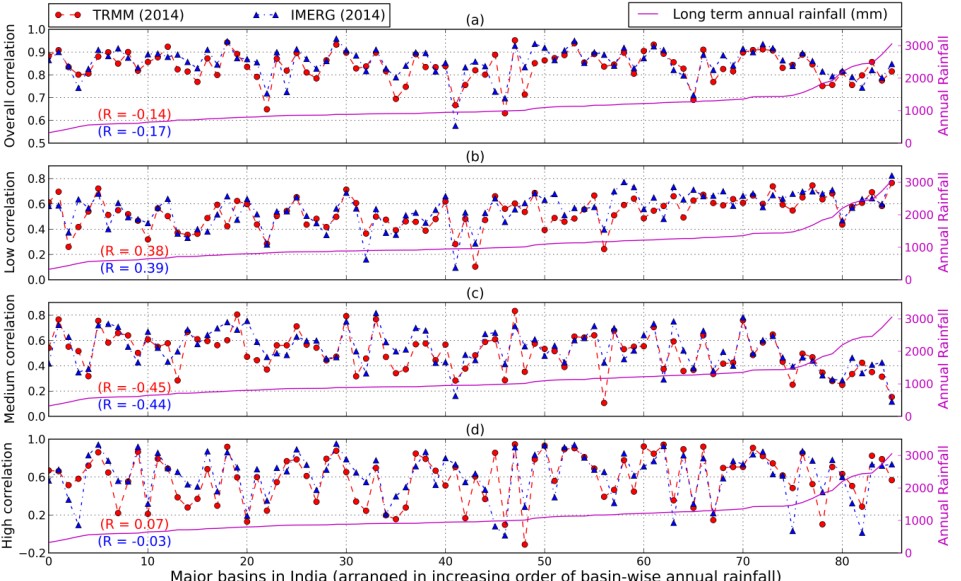

**Figure 13.** Graphical representation of correlation of IMERG (2014) and TRMM (2014) arranged in the increasing order of basin-wise average annual rainfall for **(a)** overall time series and over **(b)** low, **(c)** medium and **(d)** high rainfall regime for 86 major basins in India.

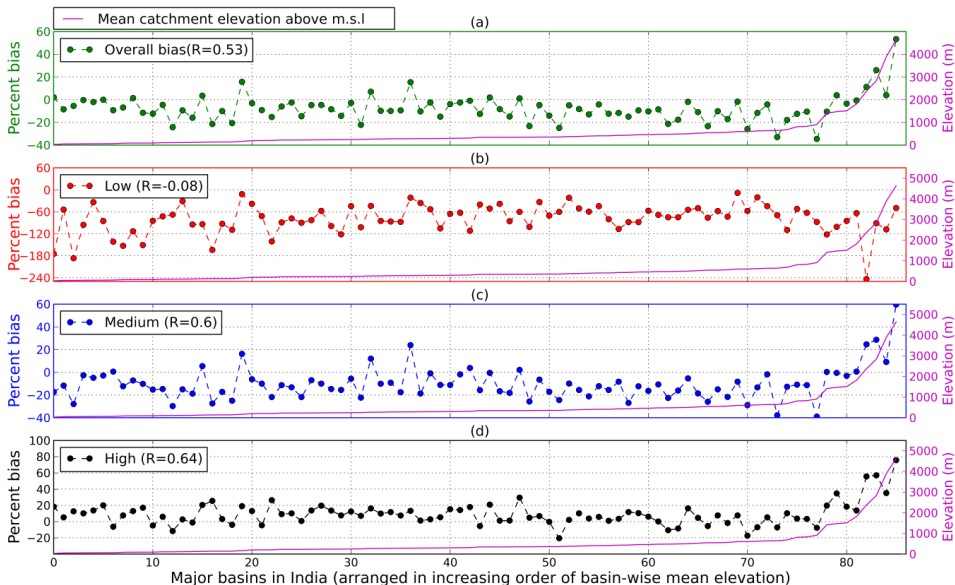

**Figure 14.** Graphical representation of percentage bias of TRMM (1998-2013) arranged in the increasing order of basin-wise average elevation over mean sea level for **(a)** overall time series and over **(b)** low, **(c)** medium and **(d)** high rainfall regime for 86 major basins in India.




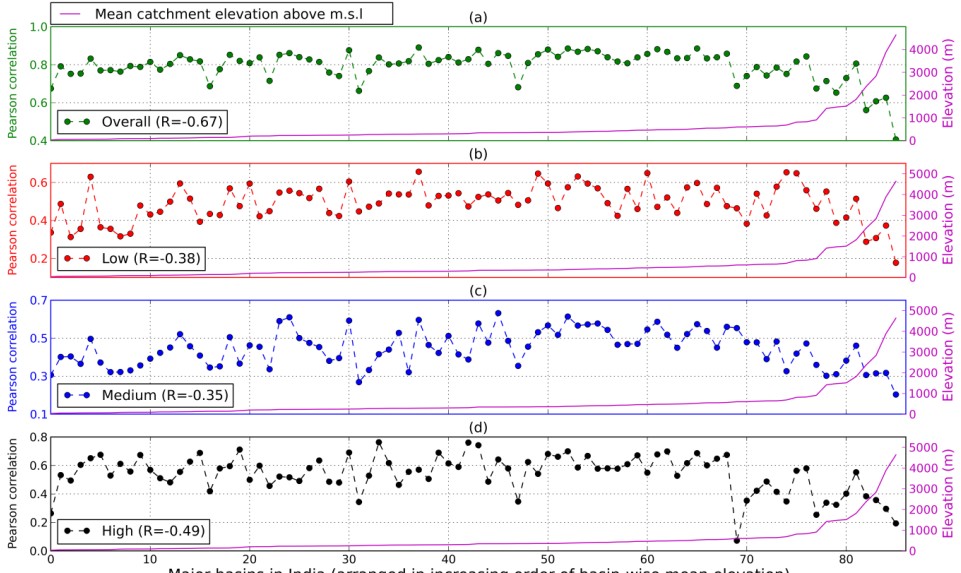

**Figure 15.** Graphical representation of correlation of TRMM (1998-2013) arranged in the increasing order of basin-wise average elevation over mean sea level for **(a)** overall time series and over **(b)** low, **(c)** medium and **(d)** high rainfall regime for 86 major basins in India.

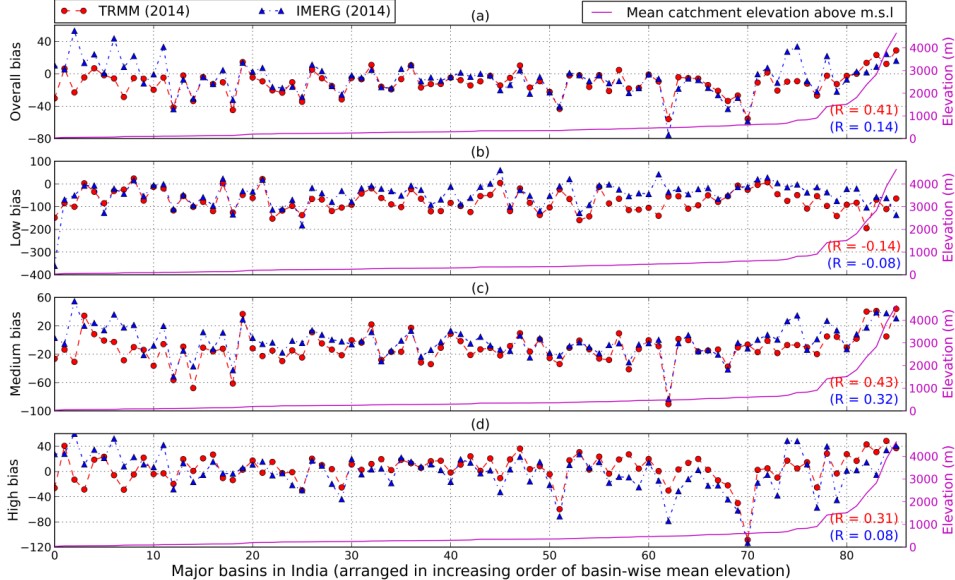

**Figure 16.** Graphical representation of percentage bias of IMERG (2014) and TRMM (2014) arranged in the increasing order of basin-wise average elevation over mean sea level for **(a)**





overall time series and over **(b)** low, **(c)** medium and **(d)** high rainfall regime for 86 major
basins in India.

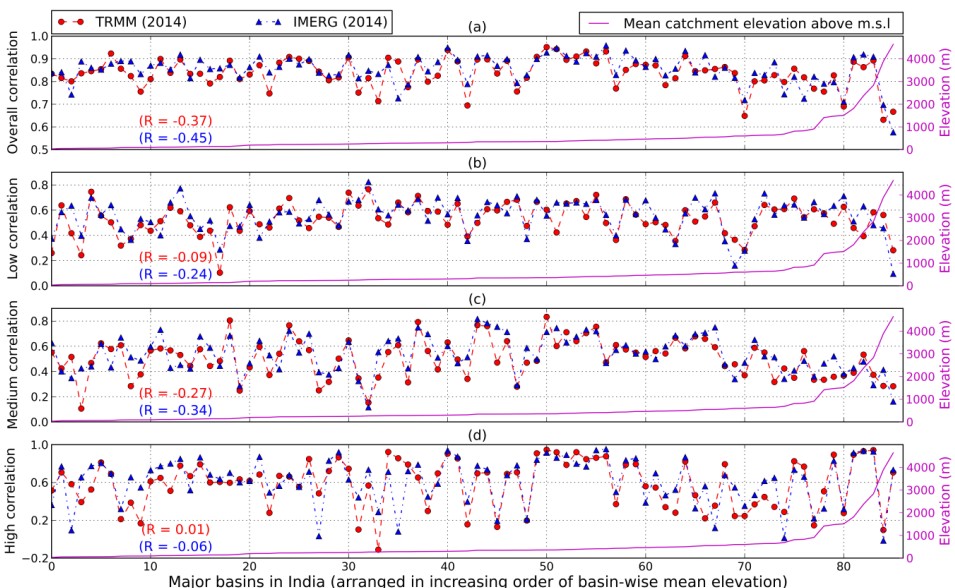


**Figure 17.** Graphical representation of correlation of IMERG (2014) and TRMM (2014)
arranged in the increasing order of basin-wise average elevation over mean sea level for **(a)**
overall time series and over **(b)** low, **(c)** medium and **(d)** high rainfall regime for 86 major
basins in India.

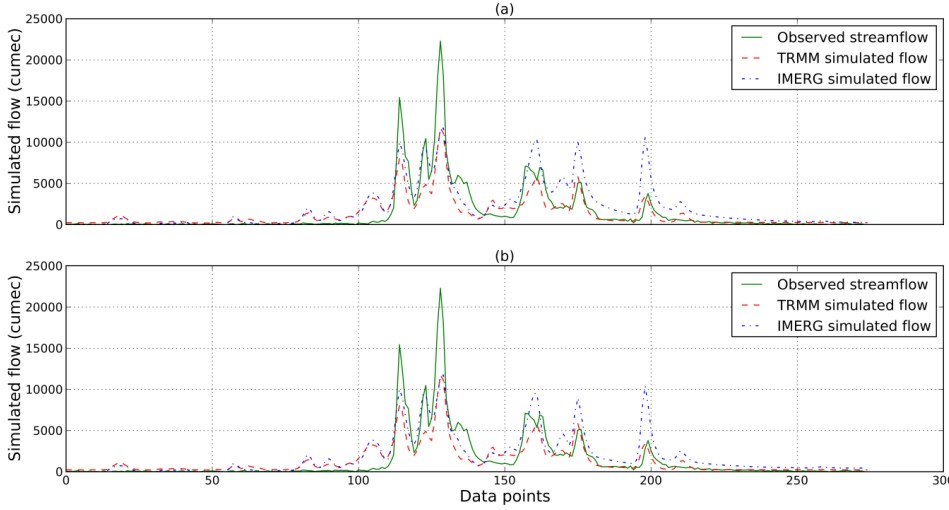






**Figure 18.** Hydrographs for TRMM and IMERG simulations (April 1, 2014 – December 31,
2014) with **(a)** IMD and **(b)** TRMM calibrated VIC model.





**Table 1.** Summary of the precipitation datasets used.

| Product name | Spatial resolution | Temporal resolution | Spatial coverage | Temporal coverage | Period used in this study |
|---|---|---|---|---|---|
| IMD Gridded Rainfall | 0.25° x 0.25° | Daily | Indian landmass | 1901-2014 | 1998-2013, 12th March, 2014 – 31st December 2014 |
| TRMM Research product | 0.25° x 0.25° | 3-hourly | 50° N-S | 1998-present | 1998-2013, 12th March, 2014 – 31st December 2014 |
| IMERG Final Run | 0.1° x 0.1° | Half-hourly | 60° N-S | 12th March, 2014 - present | 12th March, 2014 – 31st December 2014 |

**Table 2.** Contingency table used to calculate probability of detection (POD) and false alarm
ratio (FAR) at a given rainfall threshold.

| | | Simulated | |
|---|---|---|---|
| | | > Threshold | <= Threshold |
| **Observed** | > Threshold | HIT | MISS |
| | <= Threshold | FALSE | NEGATIVE |

**Table 3.** Summary of different statistical indices used to evaluate the satellite precipitation
products.

| Index | Formula | Best value | Worst value |
|---|---|---|---|
| Pearson correlation (R) | $\dfrac{\sum(X-\bar{X})(Y-\bar{Y})}{\sqrt{\sum(X-\bar{X})^2}\sqrt{\sum(Y-\bar{Y})^2}}$ | 1 | 0 |
| Percentage bias (Pbias) | $\dfrac{\sum(X-Y)}{\sum X}*100$ | 0 | $+\infty$ / $-\infty$ |
| Probability of detection (POD) | $\dfrac{HIT}{HIT+MISS}$ | 1 | 0 |
| False alarm ratio (FAR) | $\dfrac{FALSE}{HIT+FALSE}$ | 0 | 1 |
| Nash Sutcliffe efficiency (NSE) | $1-\dfrac{\sum(X-Y)^2}{\sum(X-\bar{X})^2}$ | 1 | $-\infty$ (negative value means that mean is a better estimator |





| | | | than the model). |
|---|---|---|---|
| Root mean squared error (RMSE) | $\sqrt{\dfrac{\sum(X-Y)^2}{n}}$ | 0 | $+\infty$ |

($X = Observed, \bar{X} = Observed\ mean, Y = Simulated, \bar{Y} = Simulated\ mean, n =$
$Data\ points$)
**Table 4.** Segregation of overall rainfall time series into low, medium and high rainfall time
series (R = Rainfall, μ = Mean of rainfall, σ = Standard deviation of rainfall).

| Rainfall regime | Criterion |
|---|---|
| Low | R < μ |
| Medium | R >= μ and R <= μ + 2σ |
| High | R > μ + 2σ |

**Table 5.** Performance statistics for rainfall-runoff modeling using VIC for Hirakud catchment
of Mahanadi River basin in India.

| | Time period | NSE | $R^2$ (p-value) | P-bias | RMSE (m³/s) |
|---|---|---|---|---|---|
| IMD calibration | 2000-2011 | 0.83 | 0.84 (0.01) | -16.78 | 919.88 |
| IMD validation | 2012-2014 | 0.86 | 0.88 (0.01) | -3.91 | 823.58 |
| TRMM calibration | 2000-2011 | 0.72 | 0.74 (0.01) | -18.2 | 1160.94 |
| TRMM validation | 2012-2014 | 0.73 | 0.74 (0.01) | -14 | 1128.15 |
| TRMM (IMD calibration) | 2014 | 0.72 | 0.82 (0.01) | 9.41 | 1591.09 |
| IMERG (IMD calibration) | 2014 | 0.64 | 0.68 (0.01) | -41.4 | 1786.22 |
| TRMM (TRMM calibration) | 2014 | 0.72 | 0.82 (0.01) | 9.24 | 1588.86 |
| IMERG (TRMM calibration) | 2014 | 0.7 | 0.72 (0.01) | -31.32 | 1641.82 |
