# Peer review of "Does the GPM mission improve the systematic error component in satellite"

_Hydrology and Earth System Sciences, 2016_

## Short Comment (SC1) · 11 Sep 2016

Dear Authors,

Personally, I am not very much convinced with the assumption that, the IMD gridded observed rainfall data is reliable for evaluation in entire Indian river basin. Since Rain gauges are not available in several grids during study period 2000-2014. If you see your cited reference (Pai et al ., 2014), there were about 2000-2500 gauges for all India during 2006-2010, which indicate an average rain gauge density of 0.4-0.5 gauge per $0.25°$ grid pixel. Hence, a misleading conclusion can arrive due to the errors in

observed datasets especially in terms of POD and FAR. Please justify otherwise, I would suggest you please take those grids wherever, at least one rain gauges station is available.

My second concern is about interpolation of IMERG (0.1 degree by 0.1 degree) data to 0.25 degree by 0.25 degree (as IMD resolution). How you interpolated the cases such as "if a grid is showing hit event and another adjacent grid is showing false event"/ "if a grid is showing miss event and another adjacent grid is showing false event"/ "if a grid is showing miss event and another adjacent grid is showing hit event"? Please explain

---

## Author Comment (AC1) · 15 Sep 2016

Dear Ashish,

Thank you for taking time to give your comments. We tried to address them in a point-by-point answer format:

Q: Personally, I am not very much convinced with the assumption that, the IMD gridded observed rainfall data is reliable for evaluation in entire Indian river basin. Since Rain gauges are not available in several grids during study period 2000-2014. If you see your cited reference (Pai et al., 2014), there were about 2000-2500 gauges for all India

during 2006-2010, which indicate an average rain gauge density of 0.4-0.5 gauge per 0.25 grid pixel. Hence, a misleading conclusion can arrive due to the errors in C1 observed datasets especially in terms of POD and FAR. Please justify otherwise, I would suggest you please take those grids wherever, at least one rain gauges station is available.

A: In our study, we averaged precipitation to the basin scale (91 basins over India) and carried out all the statistical analysis over basin scale, rather than grid scale. The basin scale ensures the availability of one or more rain gauges on each basin. I would like to emphasize that not only is the IMD gridded precipitation product the best estimate of large scale precipitation, but also the only one available which uses such an extensive network of rain gauges.

IMD gridded observed rainfall ($0.25°$ x $0.25°$) (referred to as IMD-R) has been quality controlled by India Meteorological Department (IMD) and a number of publications have appeared in the recent past which use this product for statistical evaluation over Indian basins (Prakash et al., 2016a, 2016b, 2016c, Shah and Mishra, 2016a, 2016b). Pai et al. (2014) mentioned that the spatial distribution of rainfall, in particular the sharp rainfall gradient from the windward to the leeward side of the Western Ghats, was very well captured in IMD-R due to the high number of rain gauge stations used in development of the gridded product. Also, the high rainfall in the North-east was well represented. In the North-most basins, the quality of rainfall data was poor due to limited number of gauging stations, thus the corresponding basins were left out from the analysis. By far, IMD-R is the best known gridded precipitation product over India.

Recent study on the statistical utility of GPM (Prakash et al., 2016a, 2016c) over India used IMD-R to evaluate the performance of GPM over TMPA. Shah and Mishra (2016b) used IMD-R to assess the utility of multiple satellite precipitation estimates in real-time streamflow monitoring over Indian sub-continental river basins. Prakash et al. (2016b) used IMD-R to compare the performance of TMPA and GSMaP for the south-west monsoon. There has been a plethora of studies using IMD-R as the reference to

evaluate the performance of multiple satellite/reanalysis precipitation products, which establishes IMD-R as a benchmark product for evaluation of satellite based rainfall estimates.

Q: My second concern is about interpolation of IMERG (0.1 degree by 0.1 degree) data to 0.25 degree by 0.25 degree (as IMD resolution). How you interpolated the cases such as "if a grid is showing hit event and another adjacent grid is showing false event"/ "if a grid is showing miss event and another adjacent grid is showing false event"/ "if a grid is showing miss event and another adjacent grid is showing hit event"? Please explain

A: We did not interpolate daily IMERG precipitation estimates from a spatial resolution of (0.1° x 0.1°) to (0.25° x 0.25°). In order to compute basin-wise precipitation, we used Thiessen Polygon method, which doesn't explicitly take care of the hit/miss statistics during interpolation. By avoiding interpolation from 0.1° to 0.25°, we ensured that high resolution rainfall information was used to compute basin precipitation. To the best knowledge of the authors, there are no commonly used interpolation methods which explicitly account for hit/miss statistics. In the climate community, people use thresholding to account for drizzle effect (Teutschbein and Seibert, 2012) which is closest to conserving threshold statistics, but that is beyond the scope of our study. For your concern about the changing frequency of hit/miss event on interpolation, this was beyond the scope of this study and maybe an interesting study for the future. Publications in the past have used simple interpolation methods to compare hit/miss statistics of GPM vs TMPA, the focus was on threshold statistics rather than the interpolation method used (Guo et al., 2016; Prakash et al., 2016c; Sahlu et al., 2016).

References:

Guo, H., Chen, S., Bao, A., Behrangi, A., Hong, Y., Ndayisaba, F., Hu, J. and Stepanian, P. M.: Early assessment of Integrated Multi-satellite Retrievals for Global Precipitation Measurement over China, Atmos. Res., 176–177, 121–133,

doi:10.1016/j.atmosres.2016.02.020, 2016.

Pai, D. S., Sridhar, L., Rajeevan, M., Sreejith, O. P., Satbhai, N. S. and Mukhopadhyay, B.: Development of a new high spatial resolution (0.25× 0.25) long period (1901–2010) daily gridded rainfall data set over India and its comparison with existing data sets over the region., Mausam, 65(1), 1–18, 2014.

Prakash, S., Mitra, A. K., AghaKouchak, A., Liu, Z., Norouzi, H. and Pai, D. S.: A preliminary assessment of GPM-based multi-satellite precipitation estimates over a monsoon dominated region, J. Hydrol., doi:10.1016/j.jhydrol.2016.01.029, 2016a.

Prakash, S., Mitra, A. K., Rajagopal, E. N. and Pai, D. S.: Assessment of TRMM-based TMPA-3B42 and GSMaP precipitation products over India for the peak southwest monsoon season, Int. J. Climatol., 36(4), 1614–1631, doi:10.1002/joc.4446, 2016b.

Prakash, S., Mitra, A. K., Pai, D. S. and AghaKouchak, A.: From TRMM to GPM: How well can heavy rainfall be detected from space?, Adv. Water Resour., 88, 1–7, doi:10.1016/j.advwatres.2015.11.008, 2016c.

Sahlu, D., Nikolopoulos, E. I., Moges, S. A., Anagnostou, E. N. and Hailu, D.: First Evaluation of the Integrated Multi-satellitE Retrieval for GPM Day-1 IMERG over the upper Blue Nile Basin, J. Hydrometeorol., doi:10.1175/JHM-D-15-0230.1, 2016.

Shah, H. L. and Mishra, V.: Hydrologic Changes in Indian Sub-Continental River Basins (1901-2012), J. Hydrometeorol., doi:10.1175/JHM-D-15-0231.1, 2016a.

Shah, H. L. and Mishra, V.: Uncertainty and Bias in Satellite-based Precipitation Estimates over Indian Sub-continental Basins: Implications for Real-time Streamflow Simulation and Flood Prediction, J. Hydrometeorol., 17(2), 615–636, doi:10.1175/JHM-D-15-0115.1, 2016b.

Teutschbein, C. and Seibert, J.: Bias correction of regional climate model simulations for hydrological climate-change impact studies: Review and evaluation of different methods, J. Hydrol., 456–457, 12–29, doi:10.1016/j.jhydrol.2012.05.052, 2012.

---

## Short Comment (SC2) · 25 Sep 2016

Dear Authors,\ I am convinced with your justification, whatever you provided.

---

## Referee Comment (RC1) · Anonymous Referee #1 · 30 Nov 2016

Review of "Does the GPM mission improve the systematic error component in satellite rainfall estimates over TRMM, an evaluation at the pan-India scale?" by Beria et al.

General Comments: This paper evaluates the performance of the new GPM satellite rainfall product compared to that of TRMM, at the basin scale for 86 basins across India. While other studies have conducted such evaluations at the grid scale, this study aims to be more relevant for hydrology, and includes a rainfall-runoff modelling exercise.

I have read the paper with interest and feel that the results provide a useful comparison

of the two precipitation products. However, I am concerned that comparisons of rainfall between the two products have been previously published for the pan-India scale, as mentioned in the introduction. Overall, I am happy with the methodology and approach, but I would have liked to see the rainfall-runoff modelling exercise completed for more than one river basin in order to set this study apart from previous similar studies, and to strengthen the conclusion that any improvements in the rainfall did not translate into improved runoff simulations. I find some descriptions within the paper confusing and feel that in almost all sections, the amount of text should be considerably reduced as this paper is very long, indeed with 9 pages before the results are presented. The results are detailed, although the authors should provide some indication of how their results compare to the studies cited in the introduction. Most of the conclusions are clear and justified, but some require clarification - I like the clear way of presenting the conclusions. The large number of figures needs to be reduced, and I have several issues with the presentation of the figures that need to be addressed. Please see the detailed comments and suggestions below.

Specific Comments & Corrections:

Text:

Title: Slight misplacement of punctuation, I believe this should read: "Does the GPM mission improve the systematic error component in satellite rainfall estimates over TRMM? An evaluation at the pan-India scale."

Lines 47-69: Interesting, and I see why this has been included, but this much detail is maybe not required as not all of these example are directly relevant to this study; this paragraph could easily be shortened.

Lines 75-77: This is almost a repeat of lines 44-46.

Lines 99-104: I would like to see more justification of the choice to focus on the basin level, to make it clear what the benefit of this study is over the previous studies the

authors have mentioned. The authors state "most" of the previous studies - what about the remaining? How does this study improve on this? Why is the basin scale more useful for water resources and policy makers? It is not clear at the moment why this would be much more useful than the grid-scale analyses.

Lines 120 & 142: I would suggest replacing the word "scanty" with "scarce", which is much more widely used and less colloquial.

Section 2.1: While background information (and especially the maps) on the study area is always appreciated, I would recommend condensing section 2.1 - not all of the information is relevant or referred to later in the paper.

Lines 201-202: This is a repeat of lines 78-79.

Line 262: Could the authors clarify this statement?

Section 3: Throughout the results section, there are a lot of statements along the lines of "IMERG outperforms TRMM in x out of y basins, but they are similar in z basins" - the authors may be able to reduce the text and number of figures by constructing a table of the number of basins in which IMERG outperforms TRMM, the number in which they are similar, and vice versa, for each skill measure evaluated in the paper. This would also be interesting for the reader to give a quick overview of these numbers without needing to read the entire text and pick them out. Of course, it is still worth discussing these and the regional differences etc. as the authors have done, but this text could be reduced.

Lines 270-275: Some of this explanation should be included in the datasets section. It is not clear why this is done like this - why were the TRMM statistics obtained for 2 periods? Also implied here is that IMERG data is only available for March - December 2014, but later in the conclusions the authors state that a longer timeseries is available. This is confusing and should be clarified. If a longer timeseries of IMERG is available, why did the authors choose to use only 2014?

[Figure]

Lines 279-280: The authors state that the two datasets show similar skills, and immediately then state that IMERG is better in 70% of the basins - this is somewhat of a contradiction.

Lines 309-310: Could the authors expand on what the implications of this result are; why is it worth noting?

Section 3.3: Throughout the section on basin-wise bias, the results are difficult to follow. Typically in the literature, a positive bias indicates over-estimation, and a negative bias indicates under-estimation. I would recommend that the authors amend the presentation of the results here to also use this convention, making it more intuitive for the reader and more consistent with the literature. This is simply a case of reversing the sign in the results, i.e. using bias = simulated - observed, instead of bias = observed - simulated. Some of the language chosen in this section is also confusing, see specific comments following:

Line 352: The authors use the term "increased" bias - it is not clear if this refers to a larger negative or positive bias.

Line 354: Surely, in the 20 basins that now exhibit a positive bias which did not before, this is indeed a decay in skill for these basins? Please clarify this statement.

Line 356: What do the authors mean by an increase in the variability of the bias? This is not clear.

Lines 354-365: The terms "lower" and "higher" when referring to bias are ambiguous; it would be better to refer to "smaller" and "larger" biases. Again, it is not clear in this paragraph whether the authors refer to positive or negative biases. Please also check the rest of the section / paper for further use of these terms.

Line 408: Does section e refer to section 3.5?

Lines 474-475: What is the reason behind this part of the evaluation? What do the authors aim to gain from this analysis? This may have been mentioned earlier in the

paper but is not completely clear and it would be good to see clarification at the start of section 3.5.

Line 488: Again, the use of "high/low" when referring to bias is confusing.

Lines 533-537: This reads as though it should be part of the methodology of the paper, rather than results.

Line 543: The term "slightly" is ambiguous - how much worse are they? How much better is the NSE? How much larger the bias?

Section 3.6: This section is presented in the introduction as a major part of the novelty of this study, but in comparison to the proportion of the paper spent discussing the rain-fall results, very little discussion is offered in terms of the hydrology. The implications of the findings are not discussed, and with only one basin used in this experiment, it is not possible to say whether the results would be similar for other basins in India or elsewhere. The aim of this experiment is left unclear and while I think it could be a very interesting part of the study, it seems somewhat unfinished. I would like to see, as the authors state would indeed be interesting, a comparison of these results for other basins in different regions in the study area.

Conclusion 1: To which parameter do the authors refer to with the quoted values?

Conclusion 3 (line 582): Do the authors refer to spatial variability? Please clarify.

Conclusion 5: Use of "higher" bias, as before.

Conclusion 7: If a longer timeseries of IMERG is available - why was this not used? This should be clarified / justified.

Lines 601-604: These statements are somewhat contradictory. The authors state throughout that IMERG outperforms TRMM in various aspects, and here state that there is a reasonable improvement, and also that the improvement is only incremental and not ground-breaking, but also that IMERG is a worthy successor of TRMM. These

statements leave the reader somewhat confused as to what the overall conclusion of the study is.

Line 611: "post forecast data assimilation scheme" - do the authors refer to post-processing?

Figures:

I'm afraid there are too many figures included in this manuscript, particularly considering the majority of figures are multi-panel. I would suggest moving a number of the figures into supplementary material. Please see below my detailed comments and recommendations for each figure:

Figure 1: Thank you for including this map, this is incredibly helpful for those readers who are not as familiar with the geography of the region. I would recommend splitting Figure 1 into two figures, one containing the two geographical maps (a) and (d), and the second comprising of (b) and (c). Also, the colour scales used for (b) and (c) are confusing - please modify these; the best way to present these would be a colour bar with just one colour for each map, ranging from light to dark with increasing values.

Figures 2.1 and 2.2. Firstly, it is strange to label two separate figures as 2.1 and 2.2 - surely these should be figures 2 and 3. Secondly, what exactly is the precipitation shown here? Is it daily precipitation? Is it averaged over a time period? This should be clarified and included on the axes / in the caption, for both 2.1 and 2.2.

Figures 3 and 5: These figures are very difficult and confusing to interpret - this data is not continuous (it represents the independent basins, rather than e.g. a continuous time period), and this is not the best way to present it. I would in fact recommend removing figures 3 and 5, and just discussing their results in the text as you have done. Figures 4 and 6, the corresponding spatial maps, are a much clearer way of presenting the data.

Figure 4: I like these maps, it is clear what they show and intuitive to interpret. However,

[Figure]

the colours used are very confusing - please amend the colour scale to use just one colour from 0 to 1 (light to dark), and avoid rainbow colours. In the case of (j), (k) and (l) it is not immediately obvious that there is a negative correlation in one or more of the basins and it is hard to spot. So on these maps, two colours should be used - the same as (a-i) for 0 to +1, with white at 0, and a different colour for the negative values. For example, the colour scale the authors have chosen for figure 8 would be perfect for figure 4, with white at 0.

Figure 6: Again, I like this figure, but the colour scale should be improved. I would recommend again a scale such as that used in figure 8, where 0 is white and the darker the colour, the larger the value. Please note that the colour scale has a big impact on the way the reader interprets the data, and incorrectly used colour scales can be misleading.

Figure 7: Again, the colour scale here is not the best option. For this data, the best would be to use one single colour, from light at 0 to dark at 1. For example, in figures (j-l), at first glance it seems that the blue basins have an opposite result to the red basins, but this is not the case.

Figure 8: While this scale would be perfect for the results shown in figure 4(j-l) and figure 6, it is not the good choice for the data presented in figure 8. As with figure 7, the best option would be one colour from light at 0 to dark at 1.

Figure 9: This graph could be removed and just discussed in the text.

figures 10 - 17: While I can see why the authors have presented the data in this way, again, there is the issue that this data is not continuous so this type of graph is not really correct, and also this is confusing for the reader. There are also a large number of similar plots here, I would suggest to pick one or two which show the most interesting results to present in the main body of the paper, and move the rest to supplementary material. Most readers would not analyse all the information in all of these figures and would appreciate the highlights, but the interested reader could easily find all the

graphs in the supplementary material. This would solve the problem of the overwhelming number of figures included in this paper. Also, I would recommend that the authors display all of these graphs (whether in the main body or the supplementary material) instead as scatter plots of the rainfall/elevation vs. bias/correlation. This would be a much more accurate and easy-to-understand way of displaying the data.

Figure 18: What is "data points"? Is it time on the x axis? Please change this label, and if it is indeed time as I suspect, please display the dates.

Recommendation: I believe that the work is of interest / useful, and warrants publication, but the manuscript indeed requires some work in terms of the descriptions and presentation of the study and its results, and clarification of some confusing aspects of the paper. Ideally, I would like to see the rainfall-runoff exercise extended.

---

## Referee Comment (RC2) · Anonymous Referee #2 · 16 Jan 2017

**Comments:**

The paper is well presented and the results reported are convincing. I have only a single concern in the methodology part regarding the evaluation of the satellite product with the gridded guage data, which is also raised by another reviewer. The inclusion of gridbox values to calculate the basin scale statistics should be strictly based on an appropriate criterion for eg, only include those grid boxes that have 2 guage values atleast (again see Hegerl et al. 2015) so that the reduction in the systematic error is not biased by any spatial infilling or interpolation techniques used in deriving the guage product. It would be useful to report these guage values per basin in a table or as a map in the supplementary document.

Introduction

Line 1: It is not just the developing part of the world, which lacks temporally and spatially continuous long-term rainfall data. Rewrite the introduction part with a few sentences in relation to the recommendations in Hegerl et al. 2015.

Reference:

Hegerl G. C., Black E., Allan R. P., Ingram W. J., Polson D., Trenberth K. E., Chadwick R. S., Arkin P. A., Balan Sarojini B., Becker A., Dai A., Durack P., Easterling D., Fowler H., Kendon E., Huffman G. J., Liu C., Marsh R., New M., Osborn T. J., Skliris N., Stott P. A., Vidale P. L., Wijffels S. E., Wilcox L. J., Willett K. & Zhang, X. Challenges in quantifying changes in the global watercycle. Bull. Am. Meteorol. Soc. doi:10.1175/bams-d-13-00212.1 (2014).

---

## Editor Comment (EC1) · H.L. Cloke (Editor) · 3 Feb 2017

Please can I encourage the authors to urgently respond to the remaining referee comments, as the discussion period has come to an end.

---

## Author Comment (AC2) · 31 Mar 2017

We thank the anonymous referee for such a detailed review. The insights provided will definitely improve the quality of the manuscript.

The referee's primary concern is regarding the hydrologic evaluation of IMERG over Indian basins. We agree that the novelty of this study lies in the hydrologic evaluation. However, the availability of streamflow data for Indian basins for the time period of IMERG data availability (starting from March 2014) is limited. WRIS, the website (http://www.india-wris.nrsc.gov.in) which provides streamflow dataset for India, is not updated and contains data for only a few gaging sites from March 2014 onwards. On going through the WRIS portal again (in January 2017) expecting better streamflow data availability, we found streamflow for the 'Barman' Gaging station in Upper Narmada basin, 'Ashti' gaging site for Wainganga river sub-basin of Lower Godavari from March 2014 apart from the gaging sites in Mahanadi basin that we have already used. We did the hydrological evaluation over Wainganga river sub-basin and will include the results in the revised manuscript. In case of Upper Narmada basin we found the flow was regulated through a reservoir and in the absence of reservoir discharge data it is extremely difficult to calibrate the model, hence we will not include it.

The referee reported the need to draw comparisons of our conclusions with those of the previous studies done for a pan-India scale. We will include them in our revised manuscript.

Another issue was regarding the length of the manuscript along with a large number of figures. The referee gave detailed suggestions, to which we have responded subsequently. We will move some figures into the supplementary and condense the text.

We have clubbed the responses to minor grammatical or language correction. To other questions pointed by the referee, we have answered them in point-by-point answer format.

*Title: Slight misplacement of punctuation, I believe this should read: "Does the GPM mission improve the systematic error component in satellite rainfall estimates over TRMM? An evaluation at the pan-India scale."*
*Lines 47-69: Interesting, and I see why this has been included, but this much detail is maybe not required as not all of these example are directly relevant to this study; this paragraph could easily be shortened.*
*Lines 75-77: This is almost a repeat of lines 44-46.*
*Lines 120 & 142: I would suggest replacing the word "scanty" with "scarce", which is much more widely used and less colloquial.*
*Section 2.1: While background information (and especially the maps) on the study area is always appreciated, I would recommend condensing section 2.1 - not all of the information is relevant or referred to later in the paper.*
*Lines 201-202: This is a repeat of lines 78-79.*
*Section 3: Throughout the results section, there are a lot of statements along the lines of "IMERG outperforms TRMM in x out of y basins, but they are similar in z basins" – the authors may be able to reduce the text and number of figures by constructing a table of the number of basins in which IMERG outperforms TRMM, the number in which they are similar, and vice versa, for each skill measure evaluated in the paper.*

*This would also be interesting for the reader to give a quick overview of these numbers without needing to read the entire text and pick them out. Of course, it is still worth discussing these and the regional differences etc. as the authors have done, but this text could be reduced.*

*Lines 279-280: The authors state that the two datasets show similar skills, and immediately then state that IMERG is better in 70% of the basins - this is somewhat of a contradiction.*

*Section 3.3: Throughout the section on basin-wise bias, the results are difficult to follow. Typically in the literature, a positive bias indicates over-estimation, and a negative bias indicates under-estimation. I would recommend that the authors amend the presentation of the results here to also use this convention, making it more intuitive for the*

*reader and more consistent with the literature. This is simply a case of reversing the sign in the results, i.e. using bias = simulated - observed, instead of bias = observed - simulated.*

*Some of the language chosen in this section is also confusing, see specific comments following:*

*Line 352: The authors use the term "increased" bias - it is not clear if this refers to a larger negative or positive bias.*

*Line 408: Does section e refer to section 3.5?*

*Line 543: The term "slightly" is ambiguous - how much worse are they? How much better is the NSE? How much larger the bias?*

We will make the suggested modifications in the revised manuscript.

**Answer to detailed comments:**

*Lines 99-104: I would like to see more justification of the choice to focus on the basin level, to make it clear what the benefit of this study is over the previous studies the authors have mentioned. The authors state "most" of the previous studies - what about the remaining? How does this study improve on this? Why is the basin scale more useful for water resources and policy makers? It is not clear at the moment why this would be much more useful than the grid-scale analyses.*

We specifically focused on basin scale because it is more relevant hydrologically. The results of a basin scale study can be directly used by the watershed managers. Most of the previous studies (as cited in the manuscript) focus on gridscale but we see a gradually changing trend to analysis on basin scale (Bisht et al., 2017; Kneis et al., 2014). It becomes easier to compare the statistical and hydrologic results when the analyses are carried out at a basin scale. Thus, we used basin scale as the reference in this manuscript.

*Line 262: Could the authors clarify this statement?*

The hydrologic model was calibrated twice, once with IMD as the rainfall forcing and once with TRMM. The model was not calibrated with IMERG as the data period was too short

(March 2014 – December 2014). Instead, the two variants of the calibrated model were validated separately using IMERG and TRMM as the rainfall forcings for the year 2014.

Regarding the warmup period, the calibration period was from 2000-2011. The year 2000 was taken as a spinup period and the results for 2000 was excluded while computing calibration statistics.

**Lines 270-275: Some of this explanation should be included in the datasets section. It is not clear why this is done like this - why were the TRMM statistics obtained for 2 periods? Also implied here is that IMERG data is only available for March – December 2014, but later in the conclusions the authors state that a longer timeseries is available. This is confusing and should be clarified. If a longer timeseries of IMERG is available, why did the authors choose to use only 2014?**

There seems to be a misunderstanding in IMERG timeseries availability. We meant to say that the IMERG is still a very young mission having started in March 2014, and as more data becomes available with time, they will lead to a clearer picture as to how IMERG compares with TRMM.

**Lines 309-310: Could the authors expand on what the implications of this result are; why is it worth noting?**

The comparison is drawn between the retrospective (1998-2013) and current (2014) time period of TRMM. Over a long period, there is a lot of temporal smoothing which may not be true for a shorter time scale. We just pointed it out in the manuscript, it doesn't really have any other significance.

**Line 354: Surely, in the 20 basins that now exhibit a positive bias which did not before, this is indeed a decay in skill for these basins? Please clarify this statement.**

As mentioned in the text, although the number of basins with positive bias increased, it wasn't a fall in skill as the basins with relatively unbiased results ( -10% <= Pbias <= 10%) increased. What really happened was some of the more negatively biased basins went to the unibased category, thus improving the overall skill.

**Line 356: What do the authors mean by an increase in the variability of the bias? This is not clear.**

This line is really ambiguous and has no clear meaning. We would remove it in the revised manuscript.

**Lines 354-365: The terms "lower" and "higher" when referring to bias are ambiguous; it would be better to refer to "smaller" and "larger" biases. Again, it is not clear in this paragraph whether the authors refer to positive or negative biases. Please also check the rest of the section / paper for further use of these terms.**

We agree that the use of "smaller" and "larger" biases instead of "lower" and "higher" biases make more sense. We would take it into account in the revised manuscript.

**Lines 474-475: What is the reason behind this part of the evaluation? What do the authors aim to gain from this analysis? This may have been mentioned earlier in the paper but is not completely clear and it would be good to see clarification at the start of section 3.5.**

We performed a correlation analysis of skill with climatology and topography to understand the systematic biases in satellite products. We will reemphasize it in section 3.5.

**Line 488: Again, the use of "high/low" when referring to bias is confusing.**

High/low bias will be changed with large/small bias.

**Lines 533-537: This reads as though it should be part of the methodology of the paper, rather than results.**

This was included to quickly recap the calibration and validation time durations. We feel this is a good practise as the reader doesn't have to go back in text and he/she can get the relevant information in brief.

**Section 3.6: This section is presented in the introduction as a major part of the novelty of this study, but in comparison to the proportion of the paper spent discussing the rainfall results, very little discussion is offered in terms of the hydrology. The implications of the findings are not discussed, and with only one basin used in this experiment, it is not possible to say whether the results would be similar for other basins in India or elsewhere. The aim of this experiment is left unclear and while I think it could be a very interesting part of the study, it seems somewhat unfinished. I would like to see, as the authors state would indeed be interesting, a comparison of these results for other basins in different regions in the study area.**

A key limitation to our study is that the hydrologic evaluation hasn't been carried out over more basins. As mentioned in the beginning of this response, there is not much that we can do about it due to the limitation in available recent streamflow data for majority of Indian basins. We will include hydrologic evaluation over Wainganga basin.

**Conclusion 1: To which parameter do the authors refer to with the quoted values?**

We referred to skill in terms of correlation. We will mention it in the revised manuscript.

**Conclusion 5: Use of "higher" bias, as before.**

We will modify the high/low bias terminology to larger/smaller in the revised manuscript.

***Conclusion 7: If a longer timeseries of IMERG is available - why was this not used? This should be clarified / justified.***

As mentioned before, there seems to be a misunderstanding in IMERG timeseries availability. We meant to say that the IMERG is still a very young mission and as more data becomes available with time, they will lead to a clearer picture as to how IMERG compares with TRMM. We will clarify that in our revised manuscript

***Lines 601-604: These statements are somewhat contradictory. The authors state throughout that IMERG outperforms TRMM in various aspects, and here state that there is a reasonable improvement, and also that the improvement is only incremental and not ground-breaking, but also that IMERG is a worthy successor of TRMM. These statements leave the reader somewhat confused as to what the overall conclusion of the study is.***

We will modify the revised manuscript to improve the clarity of this message.

***Line 611: "post forecast data assimilation scheme" - do the authors refer to postprocessing?***

We indeed meant postprocessing of streamflows.

***Figures: I'm afraid there are too many figures included in this manuscript, particularly considering the majority of figures are multi-panel. I would suggest moving a number of the figures into supplementary material. Please see below my detailed comments and recommendations for each figure:***
***Figure 1: Thank you for including this map, this is incredibly helpful for those readers who are not as familiar with the geography of the region. I would recommend splitting Figure 1 into two figures, one containing the two geographical maps (a) and (d), and the second comprising of (b) and (c). Also, the colour scales used for (b) and (c) are confusing - please modify these; the best way to present these would be a colour bar with just one colour for each map, ranging from light to dark with increasing values.***

We will split the figures as suggested by the referee.
The reason for selecting multiple colorbar for figure 1 (b) and (c) is to highlight the spatial heterogeneity in the study area. When we used a simple one colorbar, a lot of information was lost in the contrast (for instance the contrast between low rainfall in Rajasthan and medium rainfall in the Western part of Indo-Gangetic plain in figure 1(b)).

***Figures 2.1 and 2.2. Firstly, it is strange to label two separate figures as 2.1 and 2.2 - surely these should be figures 2 and 3. Secondly, what exactly is the precipitation shown here? Is it daily precipitation? Is it averaged over a time period? This should be clarified and included on the axes / in the caption, for both 2.1 and 2.2.***

We will split the figures 2.1 and 2.2 into two figures. The scatterplots show daily precipitation which we will mention in the figure description.

***Figures 3 and 5: These figures are very difficult and confusing to interpret - this data is not continuous (it represents the independent basins, rather than e.g. a continuous time period), and this is not the best way to present it. I would in fact recommend removing figures 3 and 5, and just discussing their results in the text as you have done.***

We will move the figures in supplementary material and retain the discussion in the text.

***Figures 4 and 6, the corresponding spatial maps, are a much clearer way of presenting the data.***
***Figure 4: I like these maps, it is clear what they show and intuitive to interpret. However, the colours used are very confusing - please amend the colour scale to use just one colour from 0 to 1 (light to dark), and avoid rainbow colours. In the case of (j), (k) and (l) it is not immediately obvious that there is a negative correlation in one or more of the basins and it is hard to spot. So on these maps, two colours should be used – the same as (a-i) for 0 to +1, with white at 0, and a different colour for the negative values. For example, the colour scale the authors have chosen for figure 8 would be perfect for figure 4, with white at 0.***

The rationale behind using a rainbow color to represent correlation was to focus on the spatial heterogeneity in correlation values. When we used a single color in the colorbar (ranging from light to dark), most of the spatial features of correlation were lost. For instance, it became really difficult to decipher correlation value of 0.3 from 0.6, which is rather substantial. That's why we used rainbow color bars instead of a single color bar.

***Figure 6: Again, I like this figure, but the colour scale should be improved. I would recommend again a scale such as that used in figure 8, where 0 is white and the darker the colour, the larger the value. Please note that the colour scale has a big impact on the way the reader interprets the data, and incorrectly used colour scales can be misleading.***

We will modify the color scale as recommended by the referee.

***Figure 7: Again, the colour scale here is not the best option. For this data, the best would be to use one single colour, from light at 0 to dark at 1. For example, in figures (j-l), at first glance it seems that the blue basins have an opposite result to the red basins, but this is not the case.***
***Figure 8: While this scale would be perfect for the results shown in figure 4(j-l) and figure 6, it is not the good choice for the data presented in figure 8. As with figure 7, the best option would be one colour from light at 0 to dark at 1.***

We opted for the multi color bar to highlight the spatial variability. If there is one colorbar which varies from light to dark, a lot of information is lost as the contrast decreases significantly. For instance, in the case of figure 8, FAR of anything more than 0.5 can be taken as high error, which will be lost if the colorbar had only one color scale. We will use the same color bars in figure 7 and 8 but incorporate the referee's suggestion in other figures.

***Figure 9: This graph could be removed and just discussed in the text.***

The figure will be moved to the supplementary section in the revised manuscript.

***figures 10 - 17: While I can see why the authors have presented the data in this way, again, there is the issue that this data is not continuous so this type of graph is not really correct, and also this is confusing for the reader. There are also a large number of similar plots here, I would suggest to pick one or two which show the most interesting results to present in the main body of the paper, and move the rest to supplementary material. Most readers would not analyse all the information in all of these figures and would appreciate the highlights, but the interested reader could easily find all the graphs in the supplementary material. This would solve the problem of the overwhelming number of figures included in this paper. Also, I would recommend that the authors display all of these graphs (whether in the main body or the supplementary material) instead as scatter plots of the rainfall/elevation vs. bias/correlation. This would be a much more accurate and easy-to-understand way of displaying the data.***

We will move the majority of the figures to the supplementary section. Also, we will show the respective scatterplots along with the line plots in the supplementary section.

***Figure 18: What is "data points"? Is it time on the x axis? Please change this label, and if it is indeed time as I suspect, please display the dates.***

Data points indeed here means dates. We will modify the label.

***Recommendation: I believe that the work is of interest / useful, and warrants publication, but the manuscript indeed requires some work in terms of the descriptions and presentation of the study and its results, and clarification of some confusing aspects of the paper. Ideally, I would like to see the rainfall-runoff exercise extended.***

We would like to again thank the referee for giving such a detailed feedback. We will try to incorporate the suggestions in our revised manuscript.

References:
Bisht, D. S., Chatterjee, C., Raghuwanshi, N. S. and Sridhar, V.: Spatio-temporal trends of rainfall across Indian river basins, Theor. Appl. Climatol., 1–18, doi:10.1007/s00704-017-2095-8, 2017.

Kneis, D., Chatterjee, C. and Singh, R.: Evaluation of TRMM rainfall estimates over a large Indian river basin (Mahanadi), Hydrol. Earth Syst. Sci., 18(7), 2493–2502 [online] Available from: http://www.hydrol-earth-syst-sci-discuss.net/11/1169/2014/hessd-11-1169-2014.pdf (Accessed 20 October 2014), 2014.

---

## Author Comment (AC3) · 4 Apr 2017

We thank the anonymous referee for providing valuable insights into the manuscript.

The referee's primary concern is about the influence of spatial interpolation to obtain gridded precipitation values over India, which in turn was used to obtain basin wise statistics. The gridded product was developed and quality controlled by the India Meteorological Department (IMD) (Pai et al., 2014) and has been extensively used in different statistical and hydrologic evaluations at both basin (Bisht et al., 2017; Kneis et al., 2014) and grid scale (Bisht et al., 2017; Kneis et al., 2014). As requested by the referee, we will report the number of rain gauges used to obtain the gridded precipitation product, their spatial configuration and variation with time in the revised manuscript.

We will modify the introduction section of our manuscript in line with advice from anonymous referee 1 and recommendations from Hegerl et al. (2014).

**References:**
Bisht, D. S., Chatterjee, C., Raghuwanshi, N. S. and Sridhar, V.: Spatio-temporal trends of rainfall across Indian river basins, Theor. Appl. Climatol., 1–18, doi:10.1007/s00704-017-2095-8, 2017.

Hegerl, G. C., Black, E., Allan, R. P., Ingram, W. J., Polson, D., Trenberth, K. E., Chadwick, R. S., Arkin, P. A., Sarojini, B. B., Becker, A., Dai, A., Durack, P. J., Easterling, D., Fowler, H. J., Kendon, E. J., Huffman, G. J., Liu, C., Marsh, R., New, M., Osborn, T. J., Skliris, N., Stott, P. A., Vidale, P.-L., Wijffels, S. E., Wilcox, L. J., Willett, K. M. and Zhang, X.: Challenges in Quantifying Changes in the Global Water Cycle, Bull. Am. Meteorol. Soc., 96(7), 1097–1115, doi:10.1175/BAMS-D-13-00212.1, 2014.

Kneis, D., Chatterjee, C. and Singh, R.: Evaluation of TRMM rainfall estimates over a large Indian river basin (Mahanadi), Hydrol. Earth Syst. Sci., 18(7), 2493–2502 [online] Available from: http://www.hydrol-earth-syst-sci-discuss.net/11/1169/2014/hessd-11-1169-2014.pdf, 2014.

Pai, D. S., Sridhar, L., Rajeevan, M., Sreejith, O. P., Satbhai, N. S. and Mukhopadhyay, B.: Development of a new high spatial resolution (0.25× 0.25) long period (1901–2010) daily gridded rainfall data set over India and its comparison with existing data sets over the region., Mausam, 65(1), 1–18, 2014.

Prakash, S., Mitra, A. K., Momin, I. M., Gairola, R. M., Pai, D. S., Rajagopal, E. N. and Basu, S.: A review of recent evaluations of TRMM Multisatellite Precipitation Analysis (TMPA) research products against ground-based observations over Indian land and oceanic regions, MAUSAM, 66(3), 355–366 [online] Available from: https://www.researchgate.net/profile/Satya_Prakash/publication/281115874_A_review_of_recent_evaluations_of_TRMM_Multisatellite_Precipitation_Analysis_(TMPA)_research_products_against_ground-based_observations_over_Indian_land_and_oceanic_regions/links/55e, 2015.

Prakash, S., Mitra, A. K., AghaKouchak, A., Liu, Z., Norouzi, H. and Pai, D. S.: A preliminary assessment of GPM-based multi-satellite precipitation estimates over a monsoon dominated region, J. Hydrol., doi:10.1016/j.jhydrol.2016.01.029, 2016.

Shah, H. L. and Mishra, V.: Uncertainty and Bias in Satellite-based Precipitation Estimates over Indian Sub-continental Basins: Implications for Real-time Streamflow Simulation and Flood Prediction, J. Hydrometeorol., 17(2), 615–636, doi:10.1175/JHM-D-15-0115.1, 2016.

---

## Author Response (AR1)

**Response to referee 1**

We thank the anonymous referee for such a detailed review. The insights provided definitely improved the quality of the manuscript.

The referee's primary concern was regarding the hydrologic evaluation of IMERG over Indian basins. We agree that the novelty of this study lies in the hydrologic evaluation. However, the availability of streamflow data for Indian basins for the time period of IMERG data availability (starting from March 2014) is limited. WRIS, the website (http://www.india-wris.nrsc.gov.in) which provides streamflow dataset for India, is not updated and contain data for only a few gaging sites from March 2014 onwards. On going through the WRIS portal again (in January 2017) expecting better streamflow data availability, we found streamflow for the 'Barman' Gaging station in Upper Narmada basin, 'Ashti' gaging site for Wainganga river sub-basin of Lower Godavari from March 2014 apart from the gaging sites in Mahanadi basin that we have already used. We did hydrological evaluation over Wainganga river sub-basin and included the results in the revised manuscript. In case of Upper Narmada basin we found the flow was regulated through a reservoir and in the absence of reservoir discharge data it is extremely difficult to calibrate the model, hence we did not include it.

Another issue was regarding the length of the manuscript along with a large number of figures. We reduced length of the manuscript from 10,133 words, 18 figures, 5 tables to 9141 words, 14 figures and 8 tables.

**Title: Slight misplacement of punctuation, I believe this should read: "Does the GPM mission improve the systematic error component in satellite rainfall estimates over TRMM? An evaluation at the pan-India scale"**
Title was modified as suggested.

**Lines 47-69: Interesting, and I see why this has been included, but this much detail is maybe not required as not all of these example are directly relevant to this study; this paragraph could easily be shortened.**
The text was reduced from 302 words to 228 words (lines 49-63).

**Lines 75-77: This is almost a repeat of lines 44-46.**
The line was removed.

**Lines 120 & 142: I would suggest replacing the word "scanty" with "scarce", which is much more widely used and less colloquial.**
The word "scanty" was replaced with "scarce".

**Section 2.1: While background information (and especially the maps) on the study area is always appreciated, I would recommend condensing section 2.1 - not all of the information is relevant or referred to later in the paper.**

The section was condensed from 711 to 612 words. We also incorporated description of Wainganga basin at which additional rainfall runoff exercise was carried out (lines 139-150).

**Lines 201-202: This is a repeat of lines 78-79.**
The text was removed.

**Section 3: Throughout the results section, there are a lot of statements along the lines of "IMERG outperforms TRMM in x out of y basins, but they are similar in z basins" – the authors may be able to reduce the text and number of figures by constructing a table of the number of basins in which IMERG outperforms TRMM, the number in which they are similar, and vice versa, for each skill measure evaluated in the paper. This would also be interesting for the reader to give a quick overview of these numbers without needing to read the entire text and pick them out. Of course, it is still worth discussing these and the regional differences etc. as the authors have done, but this text could be reduced.**
The results were summarized in tables 5 and 6.

**Lines 279-280: The authors state that the two datasets show similar skills, and immediately then state that IMERG is better in 70% of the basins - this is somewhat of a contradiction.**
We modified the line "Both IMERG and TRMM show quite similar skills with correlation values above 0.8, with IMERG showing better correlation in 60 out of 86 basins" to "IMERG shows better correlation in 60 out of 86 basins" (line 263-264).

**Section 3.3: Throughout the section on basin-wise bias, the results are difficult to follow. Typically in the literature, a positive bias indicates over-estimation, and a negative bias indicates under-estimation. I would recommend that the authors amend the presentation of the results here to also use this convention, making it more intuitive for the reader and more consistent with the literature. This is simply a case of reversing the sign in the results, i.e. using bias = simulated - observed, instead of bias = observed - simulated.**
The bias computation was reversed and the relevant plots and tables were modified accordingly.

**Line 352: The authors use the term "increased" bias - it is not clear if this refers to a larger negative or positive bias.**
We removed the line as it was too ambiguous.

**Line 408: Does section e refer to section 3.5?**
We modified the text as section 3.5 (line 376).

**Line 543: The term "slightly" is ambiguous - how much worse are they? How much better is the NSE? How much larger the bias?**
We included the NSE values in the description (lines 469-472).

**Answer to detailed comments:**

**Lines 99-104: I would like to see more justification of the choice to focus on the basin level, to make it clear what the benefit of this study is over the previous studies the authors have mentioned. The authors state "most" of the previous studies - what about the remaining? How does this study improve on this? Why is the basin scale more useful for water resources and policy makers? It is not clear at the moment why this would be much more useful than the grid-scale analyses.**

We specifically focused on basin scale because it is more relevant hydrologically. The results of a basin scale study can be directly used by the watershed managers. Most of the previous studies (as cited in the manuscript) focus on gridscale but we see a gradually changing trend to analysis on basin scale (Bisht et al., 2017; Kneis et al., 2014). It becomes easier to compare the statistical and hydrologic results when the analyses are carried out at a basin scale. Thus, we used basin scale as the reference in this manuscript.

**Line 262: Could the authors clarify this statement?**

The hydrologic model was calibrated twice, once with IMD as the rainfall forcing and once with TRMM. The model was not calibrated with IMERG as the data period was too short (March 2014 – December 2014). Instead, the two variants of the calibrated model were validated separately using IMERG and TRMM as the rainfall forcings for the year 2014.

Regarding the warmup period, the calibration period was from 2000-2011. The year 2000 was taken as a spinup period and the results for 2000 were excluded while computing calibration statistics.

**Lines 270-275: Some of this explanation should be included in the datasets section. It is not clear why this is done like this - why were the TRMM statistics obtained for 2 periods? Also implied here is that IMERG data is only available for March – December 2014, but later in the conclusions the authors state that a longer timeseries is available. This is confusing and should be clarified. If a longer timeseries of IMERG is available, why did the authors choose to use only 2014?**

There seems to be a misunderstanding in IMERG timeseries availability. We meant to say that the IMERG is still a very young mission having started in March 2014, and as more data becomes available with time, they will lead to a clearer picture as to how IMERG compares with TRMM.

**Lines 309-310: Could the authors expand on what the implications of this result are; why is it worth noting?**

The comparison is drawn between the retrospective (1998-2013) and current (2014) time period of TRMM. Over a long period, there is a lot of temporal smoothing which may not be true for a shorter time scale. We pointed it out in the manuscript, it doesn't have any other significance.

**Line 354: Surely, in the 20 basins that now exhibit a positive bias which did not before, this is indeed a decay in skill for these basins? Please clarify this statement.**

As mentioned in the text, although the number of basins with positive bias increased, it wasn't a fall in skill as the basins with relatively unbiased results ( -10% <= Pbias <= 10%) increased. What really happened was some of the more negatively biased basins went to the unbiased category, thus improving the overall skill.

**Line 356: What do the authors mean by an increase in the variability of the bias? This is not clear.**

We removed the line as it was too ambiguous.

**Lines 354-365: The terms "lower" and "higher" when referring to bias are ambiguous; it would be better to refer to "smaller" and "larger" biases. Again, it is not clear in this paragraph whether the authors refer to positive or negative biases. Please also check the rest of the section / paper for further use of these terms.**

We replaced the terms "lower" and "higher" biases with "smaller" and "larger" biases throughout the manuscript.

**Lines 474-475: What is the reason behind this part of the evaluation? What do the authors aim to gain from this analysis? This may have been mentioned earlier in the paper but is not completely clear and it would be good to see clarification at the start of section 3.5.**

We performed a correlation analysis of skill with climatology and topography to understand the systematic biases in satellite products. We reemphasized it in section 3.5 (lines 424-426).

**Line 488: Again, the use of "high/low" when referring to bias is confusing.**

All instances of high/low bias were changed to large/small bias.

**Lines 533-537: This reads as though it should be part of the methodology of the paper, rather than results.**

This was included to quickly recap the calibration and validation time durations. We feel this is a good practise as the reader doesn't have to go back in text and he/she can get the relevant information in brief.

**Section 3.6: This section is presented in the introduction as a major part of the novelty of this study, but in comparison to the proportion of the paper spent discussing the rainfall results, very little discussion is offered in terms of the hydrology. The implications of the findings are not discussed, and with only one basin used in this experiment, it is not possible to say whether the results would be similar for other basins in India or elsewhere. The aim of this experiment is left unclear and while I think it could be a very interesting part of the study, it seems somewhat unfinished. I would like to see, as the authors state would indeed be interesting, a comparison of these results for other basins in different regions in the study area.**

We included hydrologic evaluation over Wainganga basin of Godavari river basin.

**Conclusion 1: To which parameter do the authors refer to with the quoted values?**
We referred to skill in terms of correlation. We mention it in the revised manuscript (lines 496-498).

**Conclusion 5: Use of "higher" bias, as before.**
We modified it to "larger" bias.

**Conclusion 7: If a longer timeseries of IMERG is available - why was this not used? This should be clarified / justified.**
As mentioned before, there seems to be a misunderstanding in IMERG timeseries availability. We meant to say that the IMERG is still a very young mission and as more data becomes available with time, they will lead to a clearer picture as to how IMERG compares with TRMM. We clarified it in our revised manuscript.

**Lines 601-604: These statements are somewhat contradictory. The authors state throughout that IMERG outperforms TRMM in various aspects, and here state that there is a reasonable improvement, and also that the improvement is only incremental and not ground-breaking, but also that IMERG is a worthy successor of TRMM. These statements leave the reader somewhat confused as to what the overall conclusion of the study is.**
We will modify the text from "In essence, IMERG gives reasonable improvement in rainfall estimates across majority of the Indian basins. However, the improvement was not found to be ground breaking, rather incremental, suggesting that the GPM mission is a worthy successor of the widely acclaimed TRMM mission" to "In essence, IMERG gives reasonable improvement in rainfall estimates across majority of the Indian basins"(lines 528-529).

**Line 611: "post forecast data assimilation scheme" - do the authors refer to postprocessing?**
We indeed meant postprocessing of streamflows.

**Figure 1: Thank you for including this map, this is incredibly helpful for those readers who are not as familiar with the geography of the region. I would recommend splitting Figure 1 into two figures, one containing the two geographical maps (a) and (d), and the second comprising of (b) and (c). Also, the colour scales used for (b) and (c) are confusing - please modify these; the best way to present these would be a colour bar with just one colour for each map, ranging from light to dark with increasing values.**
We split the figures as suggested by the referee. We also included map of Wainganga catchment of the Godavari River basin (Fig. 1c)
The reason for selecting multiple colorbar for figure 1 (b) and (c) (now figs. 2a and 2b) is to highlight the spatial heterogeneity in the study area. When we used a simple one colorbar, a lot of information was lost in the contrast (for instance the contrast between low rainfall in Rajasthan and medium rainfall in the Western part of Indo-Gangetic plain in figure 1(b)).

**Figures 2.1 and 2.2. Firstly, it is strange to label two separate figures as 2.1 and 2.2 - surely these should be figures 2 and 3. Secondly, what exactly is the precipitation shown here? Is it daily precipitation? Is it averaged over a time period? This should be clarified and included on the axes / in the caption, for both 2.1 and 2.2.**

We split the figures 2.1 and 2.2 into two figures (Figs. 3 and 4). The scatterplots show daily precipitation which we mentioned in the figure descriptions.

**Figures 3 and 5: These figures are very difficult and confusing to interpret - this data is not continuous (it represents the independent basins, rather than e.g. a continuous time period), and this is not the best way to present it. I would in fact recommend removing figures 3 and 5, and just discussing their results in the text as you have done.**

The figures were moved to supplementary material (Figs S1, S2).

**Figures 4 and 6, the corresponding spatial maps, are a much clearer way of presenting the data.**
**Figure 4: I like these maps, it is clear what they show and intuitive to interpret. However, the colours used are very confusing - please amend the colour scale to use just one colour from 0 to 1 (light to dark), and avoid rainbow colours. In the case of (j), (k) and (l) it is not immediately obvious that there is a negative correlation in one or more of the basins and it is hard to spot. So on these maps, two colours should be used – the same as (a-i) for 0 to +1, with white at 0, and a different colour for the negative values. For example, the colour scale the authors have chosen for figure 8 would be perfect for figure 4, with white at 0.**

The rationale behind using a rainbow color to represent correlation was to focus on the spatial heterogeneity in correlation values. When we used a single color in the colorbar (ranging from light to dark), most of the spatial features of correlation were lost. For instance, it became really difficult to decipher correlation value of 0.3 from 0.6, which is rather substantial. That's why we used rainbow color bars instead of a single color bar. In the revised manuscript, fig. 4 is moved to fig. 5.

**Figure 6: Again, I like this figure, but the colour scale should be improved. I would recommend again a scale such as that used in figure 8, where 0 is white and the darker the colour, the larger the value. Please note that the colour scale has a big impact on the way the reader interprets the data, and incorrectly used colour scales can be misleading.**

We modified the color scale as recommended by the referee (Fig. 6).

**Figure 7: Again, the colour scale here is not the best option. For this data, the best would be to use one single colour, from light at 0 to dark at 1. For example, in figures (j-l), at first glance it seems that the blue basins have an opposite result to the red basins, but this is not the case.**

**Figure 8: While this scale would be perfect for the results shown in figure 4(j-l) and figure 6, it is not the good choice for the data presented in figure 8. As with figure 7, the best option would be one colour from light at 0 to dark at 1.**

We opted for the multi color bar to highlight the spatial variability. If there is one colorbar which varies from light to dark, a lot of information is lost as the contrast decreases significantly. For instance, in the case of figure 8, FAR of anything more than 0.5 can be taken as high error, which will be lost if the colorbar had only one color scale. We will use the same color bars in figure 7 and 8 but incorporate the referee's suggestion in other figures.

**Figure 9: This graph could be removed and just discussed in the text.**

The figure was moved to the supplementary section (Fig. S3).

**figures 10 - 17: While I can see why the authors have presented the data in this way, again, there is the issue that this data is not continuous so this type of graph is not really correct, and also this is confusing for the reader. There are also a large number of similar plots here, I would suggest to pick one or two which show the most interesting results to present in the main body of the paper, and move the rest to supplementary material. Most readers would not analyse all the information in all of these figures and would appreciate the highlights, but the interested reader could easily find all the graphs in the supplementary material. This would solve the problem of the overwhelming number of figures included in this paper. Also, I would recommend that the authors display all of these graphs (whether in the main body or the supplementary material) instead as scatter plots of the rainfall/elevation vs. bias/correlation. This would be a much more accurate and easy-to-understand way of displaying the data.**

We moved the four figures related to climatology to supplementary and preserved the ones related to topography as they showed more significant results. We couldn't show the scatterplots because the scales are not standardized. If we do a scatterplot of correlation with elevation, correlation varies from -1 to 1 but elevation varies from 0 to 4500m and there is no meaningful 1:1 line to draw any inference. Hence we stuck with the line plots. However, we removed the lines and only kept the points as then it is easier to see if there is a cluster of points behaving in a certain way. The modified plots are Figs. 9-12.

**Figure 18: What is "data points"? Is it time on the x axis? Please change this label, and if it is indeed time as I suspect, please display the dates.**

We will modify the label to "Number of days since April 1st, 2014.

**Response to referee 2**

We thank the anonymous referee for providing valuable insights into the manuscript.

The referee's primary concern is about the influence of spatial interpolation to obtain gridded precipitation values over India, which in turn was used to obtain basin wise statistics. The gridded product was developed and quality controlled by the India Meteorological Department (IMD) (Pai et al., 2014) and has been extensively used in different statistical and hydrologic evaluations at both basin (Bisht et al., 2017; Kneis et al., 2014) and grid scale (Bisht et al., 2017; Kneis et al., 2014). As requested by the referee, we reported the number of rain gauges used to obtain the gridded precipitation product, their spatial configuration and variation with time in the supplementary section of the manuscript (Figs. S8, S9) and mentioned it in the section on IMD dataset (section 2.2.1). The description is below.

**Station related info**

Gridded rainfall product of IMD is prepared from station record of rainfall. However, the total number of stations used varies from year to year, the reasons may be attributed to maintenance, cost of operation, data quality, and man power availability. Figure S8 shows the maximum total number of stations used for preparing the high resolution gridded rainfall product during 1998-2014. Decline in the number of station over the period of time is evident form the fig. S8, nevertheless, the IMD gridded rainfall product has been widely used by the researchers in similar studies as discussed in the manuscript. Spatial distribution of rainfall stations during 1998-2014 at 0.25 degree spatial resolution is shown in the fig. S9, to reduce the number of plots maps are shown at 3 year interval during 1998-2014. Comparatively, a high density of rainfall station network can be seen in southern peninsular India. In some cases, the total number of stations in a single grid can go up to 11.

[Figure]

**Fig S8.** Total maximum number of active rainfall stations across all the grids during 1998-2014

[Figure]

**Fig S9.** Spatial distribution of rainfall station during 1998, 2002, 2006, 2010, and 2014

[revised manuscript text omitted]

---

## Referee Report (RR1)

**Review of "Does the GPM mission improve the systematic error component in satellite rainfall estimates over TRMM? An evaluation at the pan-India scale" by Beria et al.**

**General Comments**

Upon reading the revised manuscript by Beria et al., "Does the GPM mission improve the systematic error component in satellite rainfall estimates over TRMM? An evaluation at the pan-India Scale.", I find it to be much improved, with interesting added analysis including rainfall-runoff simulations over a second basin, many of the previously unclear points clarified, and the visualisation of the figures improved. Overall, the manuscript reads well, but I recommend some further minor revisions and/or clarifications of the text and figures below. Additionally, there are still too many figures and tables included in the main manuscript; I have further suggested some figures that should be moved to the supplementary material. While the results are interesting and useful, it would be beneficial to see more discussion of the implications of the results.

**Specific Comments & Corrections**

Line 34: Surely also, in addition to / regardless of climate change, flooding in itself is a current threat that this paper is relevant to.

Line 70: TMPA acronym is first expanded at line 181, please expand here at first use

Line 103: this should be updated to "two flood prone basins", in this revision

Line 223-224: Do the authors have a reference for the Thiessen Polygon method?

Line 419-420: Lots of false alarms at very high rainfall thresholds implies that it is not good at capturing the extremes; this could have implications for flood modelling etc. and I expected to see this mentioned in the discussions / conclusions, particularly as the authors go on to complete a rainfall-runoff modelling exercise and find that the peak flows are not captured well. This should be an interesting point to mention.

Lines 477-478: The authors refer to a negative bias showing overprediction - it is not clear if underprediction is really what is meant, as the bias is negative. Please clarify in the text, or check that you have positive biases for overprediction, and negative biases for underprediction, to be consistent with the biases presented earlier in the paper.

Conclusion 7: Looking at the hydrographs, the results with IMERG and TRMM are pretty similar regardless of the calibration, for both basins. Neither are capable of capturing the peak flows, despite the results finding that precipitation is generally improved in IMERG. Could the problem be more due to the hydrological model used (would a different model perhaps result in better prediction of the peaks using either of the rainfall datasets?) rather than the choice between TRMM or IMERG? Or is it the case that the rainfall datasets cannot capture the extreme rainfall? A possible limitation that could be interesting to mention.

Line 536: 'Post forecast data assimilation scheme' - do the authors refer to postprocessing?

Figure 5: From the authors' response, I accept that the contrast may be hard to see using a colour scale with only 1 colour. I still find this figure very hard to interpret. Perhaps it would be possible to use cool colours (blue to pink) for positive correlation, and warm (orange and red) for negative correlation? This would allow the use of more colours to avoid the contrast issue, would allow the basins with negative correlation to stand out further making the plot easier to interpret, and would avoid the use of green and red on the same figure (which it is generally recommended to avoid, due to the % of the population who are colourblind).

Figures 7 & 8: Indeed, I agree with the authors comment regarding the use of two colours, I had not realised this point about the FAR > 0.5. Perhaps the relevance of FAR > 0.5 / POD < 0.5 could be mentioned in table 3.

Figures 9-12: These plots are discussed briefly in the manuscript; while the results are interesting, I don't think the discussion warrants four 4-panel figures. The results presented in section 3.5 are clear without needing to refer to the figures, and I would recommend moving them to the supplementary material.

Figures S1 - S3: The points on these figures should not be joined with a continuous line, as these are not continuous data; this can be difficult to interpret and can be misleading.

---

## Author Response (AR2)

**Response to referee**

We thank the anonymous referee for the suggestions and we are happy to say that we have been able to incorporate most of them. The number of figures have been reduced from 14 to 10 in the revised manuscript. Additionally, we proof read the final manuscript and made some slight modifications to the text in order to make it clearer. All the modifications were done using track changes. A copy of the track changed document is uploaded along with the final manuscript. Point-by-point answers to the referee's comments are below.

**Line 34: Surely also, in addition to / regardless of climate change, flooding in itself is a current threat that this paper is relevant to.**

That is absolutely correct. Including the climate change aspect in the description implies that the situation of floods may worsen in the future, and satellite based precipitation estimates can be the solution for real-time flood forecasting. The new line reads as:
*With the threats of climate change looming large, high quality precipitation products (in terms of accuracy, spatial and temporal resolution) are the need of the hour to analyse hydro-meteorological processes in real time.*

**Line 70: TMPA acronym is first expanded at line 181, please expand here at first use**

TMPA acronym has been expanded in line 70.

**Line 103: this should be updated to "two flood prone basins", in this revision**

We revised the text to clearly state two flood prone basins along with their names (line 103-107). The revised line reads:
*Finally, we used a macroscale hydrologic model (Variable Infiltration Capacity (VIC)) to evaluate TRMM and IMERG over two flood prone basins in Eastern India (Hirakud catchment of the Mahanadi River basin and Wainganga catchment of the Godavari River basin) for the year 2014.*

**Line 223-224: Do the authors have a reference for the Thiessen Polygon method?**

We added a reference (Schumann, 1998) for Thiessen Polygon method (line 224-226 in revised manuscript).

**Line 419-420: Lots of false alarms at very high rainfall thresholds implies that it is not good at capturing the extremes; this could have implications for flood modelling etc. and I expected to see this mentioned in the discussions / conclusions, particularly as the authors go on to complete a rainfall-runoff modelling exercise and find that the peak flows are not captured well. This should be an interesting point to mention.**

We mentioned the point in conclusion #4 in the revised manuscript (line 513-518). The line reads as below:
*At very high rainfall thresholds (>95 percentile), TRMM exhibited high false alarm ratio (FAR), especially in the North-eastern and Southern basins, implying that they do not capture the extreme precipitation magnitudes well. This was also seen in the rainfall-runoff exercise*

*where the peak flows were underpredicted in Mahanadi and Wainganga River basins, both in the case of TRMM and IMERG.*

**Lines 477-478: The authors refer to a negative bias showing overprediction - it is not clear if underprediction is really what is meant, as the bias is negative. Please clarify in the text, or check that you have positive biases for overprediction, and negative biases for underprediction, to be consistent with the biases presented earlier in the paper.**
This was a typo. We changed "overprediction" to "underprediction" in the revised manuscript (line 480). Thank you for pointing it out.

**Conclusion 7: Looking at the hydrographs, the results with IMERG and TRMM are pretty similar regardless of the calibration, for both basins. Neither are capable of capturing the peak flows, despite the results finding that precipitation is generally improved in IMERG. Could the problem be more due to the hydrological model used (would a different model perhaps result in better prediction of the peaks using either of the rainfall datasets?) rather than the choice between TRMM or IMERG? Or is it the case that the rainfall datasets cannot capture the extreme rainfall? A possible limitation that could be interesting to mention.**
Thank you for pointing this out. We added the following text to point #7 of conclusion (line 534-539).
*It will also be useful to see if other hydrologic models can capture peak flows more accurately when forced with TRMM/IMERG in Mahanadi and Wainganga basins. This would mean that the poor representation of peak flows is a function of model structural uncertainty, and not the satellite precipitation products driving the model. This will make a very interesting future case study.*

**Line 536: 'Post forecast data assimilation scheme' - do the authors refer to postprocessing?**
Yes, we refer to postprocessing. We clarified it in the revised manuscript and included a reference (Ye et al., 2014) for it (line 546-549).

**Figure 5: From the authors' response, I accept that the contrast may be hard to see using a colour scale with only 1 colour. I still find this figure very hard to interpret. Perhaps it would be possible to use cool colours (blue to pink) for positive correlation, and warm (orange and red) for negative correlation? This would allow the use of more colours to avoid the contrast issue, would allow the basins with negative correlation to stand out further making the plot easier to interpret, and would avoid the use of green and red on the same figure (which it is generally recommended to avoid, due to the % of the population who are colourblind).**
We agree that the figure is not best suited for color blind people. As advised by the reviewer, we revised the colorbar of figure 5, using warm colors (variants of brown) for low correlation and cool colors (variants of blue). The new figure is placed below for illustration.

[Figure]

**Figure 5**. Spatial representation of correlation of TRMM (1998-2013), TRMM (2014) and IMERG (2014) over 86 delineated river basins across India for **(a) – (c)** overall time series, **(d) – (f)** low, **(g) – (i)** medium and **(j) – (l)** high rainfall regime.

**Figures 7 & 8: Indeed, I agree with the authors comment regarding the use of two colours, I had not realised this point about the FAR > 0.5. Perhaps the relevance of FAR > 0.5 / POD < 0.5 could be mentioned in table 3.**

We did not mention the significance of FAR < 0.5 or POD > 0.5 in table 3 as defining 0.5 as a threshold for low or high POD/FAR seems arbitrary.

**Figures 9-12: These plots are discussed briefly in the manuscript; while the results are interesting, I don't think the discussion warrants four 4-panel figures. The results presented in section 3.5 are clear without needing to refer to the figures, and I would recommend moving them to the supplementary material.**
Figures 9-12 have been moved to supplementary material (Figs. S4-S7).

**Figures S1 - S3: The points on these figures should not be joined with a continuous line, as these are not continuous data; this can be difficult to interpret and can be misleading.**
Continuous lines have been removed from figures S1-S3 in the revised supplementary material.

**References:**

[revised manuscript text omitted]